# Ras/MAPK signalling intensity defines subclonal fitness in a mouse model of hepatocellular carcinoma

**Anthony Lozano[1], Francois-Régis Souche[1,2], Carine Chavey[1], Valérie Dardalhon[1], Christel Ramirez[3], Serena Vegna[3], Guillaume Desandre[1], Anaïs Riviere[1], Amal Zine El Aabidine[1], Philippe Fort[4], Leila Akkari[3], Urszula Hibner[1*†], Damien Grégoire[1*†]**

[1]Institut de Génétique Moléculaire de Montpellier, University of Montpellier, Montpellier, France; [2]Department of surgery and liver transplantation, Hopital Saint Eloi Hopitaux universitaires de Montpelier, Montpellier, France; [3]Division of Tumor Biology and Immunology, Netherlands Cancer Institute, Oncode Institute, Amsterdam, Netherlands; [4]Centre de Recherche en Biologie Cellulaire de Montpellier (CRBM), University of Montpellier, CNRS, Montpellier, France

**\*For correspondence:**
ula.hibner@igmm.cnrs.fr (UH);
damien.gregoire@igmm.cnrs.
fr (DG)

†These authors contributed equally to this work

**Abstract** Quantitative differences in signal transduction are to date an understudied feature of tumour heterogeneity. The MAPK Erk pathway, which is activated in a large proportion of human tumours, is a prototypic example of distinct cell fates being driven by signal intensity. We have used primary hepatocyte precursors transformed with different dosages of an oncogenic form of Ras to model subclonal variations in MAPK signalling. Orthotopic allografts of Ras-transformed cells in immunocompromised mice gave rise to fast-growing aggressive tumours, both at the primary location and in the peritoneal cavity. Fluorescent labelling of cells expressing different oncogene levels, and consequently varying levels of MAPK Erk activation, highlighted the selection processes operating at the two sites of tumour growth. Indeed, significantly higher Ras expression was observed in primary as compared to secondary, metastatic sites, despite the apparent evolutionary trade-off of increased apoptotic death in the liver that correlated with high Ras dosage. Analysis of the immune tumour microenvironment at the two locations suggests that fast peritoneal tumour growth in the immunocompromised setting is abrogated in immunocompetent animals due to efficient antigen presentation by peritoneal dendritic cells. Furthermore, our data indicate that, in contrast to the metastatic-like outgrowth, strong MAPK signalling is required in the primary liver tumours to resist elimination by NK (natural killer) cells. Overall, this study describes a quantitative aspect of tumour heterogeneity and points to a potential vulnerability of a subtype of hepatocellular carcinoma as a function of MAPK Erk signalling intensity.

## Editor's evaluation

This is an important study that demonstrates a role for elevated Ras signaling in the growth of liver cancer in immunocompetent mice that is mediated, in part, by resistance to NK-mediated killing. The conclusions are, in large part, supported by solid data. Further studies are required to identify molecular mechanisms.

## Introduction

Tumour subclonal heterogeneity is defined by cancer cells' intrinsic properties and by phenotypic adaptations to extrinsic signals from the microenvironment (*Marusyk et al., 2020*). Specific environments are shaped by the physiological constraints imposed by the tumour localization and by the dynamic interactions with the growing tumour (*Gerstung et al., 2020*). The environment impacts the tumour phenotype, first by triggering adaptive responses from the highly plastic cancer cells (*Pastushenko and Blanpain, 2019*) and second, by exerting selective pressures, leading to expansion of some, and elimination of other, tumoural subclones (*Janiszewska et al., 2019*). While the notion of inter- and intra-tumour heterogeneities and their consequences for personalized medicine have gained a strong momentum in the last decade (*Losic et al., 2020*; *Molina-Sánchez et al., 2020*), quantitative aspects leading to differences in the intensity of oncogenic signal transduction within the selected, clonal tumour cell populations (see e.g. *Gross et al., 2019*) are so far understudied.

Genomic analyses identified major oncogenic pathways for numerous tumour types (*Sanchez-Vega et al., 2018*). For example, hepatocellular carcinoma (HCC) has been classified into several classes, defined by their genomic and physiopathological characteristics (*Llovet et al., 2021*). While some oncogenic driver mutations fall neatly in such defined categories, for example, mutations that activate the β-catenin pathway or that inactivate the p53 tumour suppressor (*Zucman-Rossi et al., 2015*), others are frequently present in several different HCC types. An interesting example is the activation of the Ras/MAPK Erk pathway. Although mutations of Ras GTPases are found in only 2–4% of HCC patients, multiple activators and regulators of the pathway, such as FGF19, RSK2, RASAL1, RASSF1, and DUSPs, are more frequently mutated. Altogether, the Ras/MAPK is estimated to be activated in 40% of HCC (*Delire and Stärkel, 2015*; *Lim et al., 2018*).

Activation of Ras/MAPK Erk signalling is an essential feature of proliferating cells (*Pagès et al., 1993*) and the genetic abrogation of several of the components of the pathway gives rise to embryonic lethality due to a decrease in signal intensity (*Dorard et al., 2017*). Moreover, variations in signal intensity, duration, and tissue-specific context give rise to distinct cellular outcomes (*Lenormand et al., 1993*; *Marshall, 1995*; *Pouysségur et al., 2002*).

In this work, we transformed hepatocyte precursors (bipotent mouse embryonic liver [BMEL] cells) with different number of copies of an oncogenic mutant Ras (H-Ras$^{G12V}$), which entailed variable intensity of signalling through the Ras/MAPK Erk pathway. We show that these cells tolerate a wide range of activated Ras dosage in vitro. In contrast, in vivo tumours display a much narrower range of active Ras levels and the consequent Erk signalling, likely attributable to selective pressures from the microenvironment. Specifically, cells that sustain low levels of Ras/MAPK Erk pathway activation are gradually eliminated from the liver primary tumours. Strikingly, they persist in secondary tumours that develop in the peritoneal cavity. Thus, specific microenvironments present in primary and in metastatic tumour locations, which differ in their macrophage, dendritic, and natural killer (NK) cell immune landscape, select distinct tumoural subclones that are characterized by site-specific Ras signalling dosage.

## Results

### The phenotype of hepatic progenitors is affected by the Ras$^{G12V}$ gene dosage

In order to investigate the consequences of distinct Ras oncogenic dosage on tumour development, we used BMEL, bipotential precursors of the two epithelial hepatic lineages, derived from embryonic mouse liver (*Akkari et al., 2010*; *Strick-Marchand and Weiss, 2003*; *Strick-Marchand and Weiss, 2002*). BMEL are not transformed and retain major characteristics of primary hepatic progenitors, including the capacity to repopulate a damaged adult liver (*Strick-Marchand et al., 2004*). We have previously reported that the expression of an oncogenic form of Ras (H-Ras$^{G12V}$) is sufficient to transform a subset of BMEL clones (*Akkari et al., 2012*; *Bacevic et al., 2019*), including the ones used for the current study. These cells express p19ARF and wild-type p53, but not p16-INK4A (*Figure 1— figure supplement 1*), which could account for their escape from Ras-induced senescence (*Xue et al., 2007*).

Because cellular phenotypes are sensitive to the Ras oncogene dosage (*Kerr et al., 2016*; *Mueller et al., 2018*) and the signalling intensity of its downstream effector, the MAPK Erk pathway (*Dikic et al., 1994*; *Traverse et al., 1994*), we asked whether a specific level of Ras$^{G12V}$ expression was

required for hepatic tumour growth in vivo. To obtain cellular populations with distinct levels of Ras[G12V] expression, BMEL cells were transduced with bicistronic lentiviral vectors encoding H-Ras[G12V] and either a Venus or a mCherry fluorescent protein (*Figure 1A* upper). Cells were sorted by flow cytometry on the basis of the fluorescence intensity, giving rise to populations that we named 'Ras[LOW]' and 'Ras[HIGH]' (*Figure 1A* middle). Since the parental BMEL cells used in this study were clonally derived, the BMEL-Ras populations were isogenic, except for the copy number and insertion sites of the Ras[G12V] – Venus/mCherry transgene and the consequent level of their expression. As expected, the expression levels of the fluorescent markers faithfully reflected the Ras oncogene mRNA levels (*Figure 1—figure supplement 2*) and the mean levels of Ras[G12V] protein were different in the Ras[LOW] versus the Ras[HIGH] populations, as shown for Ras[LOW-Cherry] and Ras[HIGH-Venus] in *Figure 1A*. However, the Ras[LOW] and Ras[HIGH] cells, while respectively enriched in weak and strong H-Ras[G12V] expressors, remained quite heterogeneous with respect to the oncogene expression. Indeed, because of the gates used for sorting, the distribution of Ras expression levels presented an overlap between the two populations (*Figure 1* and *Figure 1—figure supplement 2*). Importantly, neither the level of Ras expression nor the nature of the co-expressed fluorescent protein altered the in vitro proliferation rate of the transformed BMEL cells (*Figure 1—figure supplement 3*).

While the parental BMEL cells did not form colonies when deprived of anchorage, Ras-transformed cells grew in soft agar, with the Ras[HIGH] cells giving significantly more colonies than their Ras[LOW] counterparts in this surrogate assay of transformation (*Figure 1B*). Several major signal transduction pathways lay downstream of Ras activation. To gain insight into their activation status in H-Ras[G12V]-transformed BMEL cells, we assayed phosphorylation levels of a set of kinases (*Figure 1C*). Oncogenic Ras had no effect on AKT[S473] and the GSKb[S9], two downstream targets of PI(3)K that were strongly phosphorylated both in the parental and in the Ras-transformed cells. Similarly, no differences were detected for JNK and p38 pathway components. In contrast, although this assay did not reveal increased MEK or Erk activation, Ras signalling gave rise to a strong increase in the phosphorylation of Rsk 1 and 2, which are downstream targets of MAPK Erk signalling. Moreover, while the proteomic array assay was not sufficiently sensitive to distinguish between the Ras[LOW] and Ras[HIGH] signalling, the oncogenic dosage did translate into differences of the Ras/MAPK Erk signalling, as evidenced by distinct transcriptional signatures of known Erk delayed early target genes (*Brant et al., 2017*; *Figure 1D*).

In order to further investigate how the intensity of Ras signalling governs the hepatic transcriptional programmes, we performed RNAseq analysis of parental BMEL, Ras[LOW] and Ras[HIGH] cells in culture. Setting the thresholds at fold change log2=1.5 and p-value <0.01 identified over 1000 genes modulated by Ras[G12V] signalling in these hepatic progenitor cells (*Figure 1E*). The expression of 160 genes was modulated by Ras independently of its mean expression level in the tested population (*Figure 1E* left, *Supplementary file 1a*), thus representing a gene signature for the low threshold Ras/MAPK signalling. In a second group of 65 genes, target gene expression was modified only in the Ras[HIGH] cells (*Figure 1E* middle, *Supplementary file 1b*). For these genes, a high-intensity Ras signalling is required for the mRNA accumulation. Finally, the third group contains the genes expression of which correlated with the mean level of Ras expression. We identified six genes (*Al467606, Aim2, Dynap, Htra3, Itgb7, Tspan13*) whose expression was gradually increased in Ras[LOW] and Ras[HIGH] cell populations (*Figure 1E* right, *Supplementary file 1c*). Similarly, we found nine genes (*Ppp2r2b, Cbr1, Pmp22, Ptp4a3, Pmaip1, Thbs1, Akap12, Sulf2, Crip2*) the expression of which was gradually repressed. Thus, our analysis identifies novel sets of quantitatively regulated Ras/MAPK target genes in hepatic cells. In the general framework of questioning the impact of Ras oncogenic dosage on the tumour-stroma interactions, we note the presence of genes in the gene ontology (GO) category of immune response regulation (*Figure 1F*).

## Specific Ras[G12V] gene dosage is selected during tumour growth

Orthotopic injections of 10[5] BMEL-Ras cells into immunocompromised recipients systematically gave rise within 3–4 weeks to hepatocellular tumours at the site of injection as well as to frequent extrahepatic growth in the peritoneal cavity. First, we concentrated on primary tumours arising in the liver. Allografts of Ras[LOW] and Ras[HIGH] cells both produced poorly differentiated and fast growing, aggressive tumours, as witnessed by their overall size, high proliferation index, and typically a multi-nodular morphology (*Figure 2A*). In accordance with their superior capacity for anchorage-independent growth (*Figure 1B*), the Ras[HIGH] cells gave rise to significantly larger tumours than their Ras[LOW] counterparts,

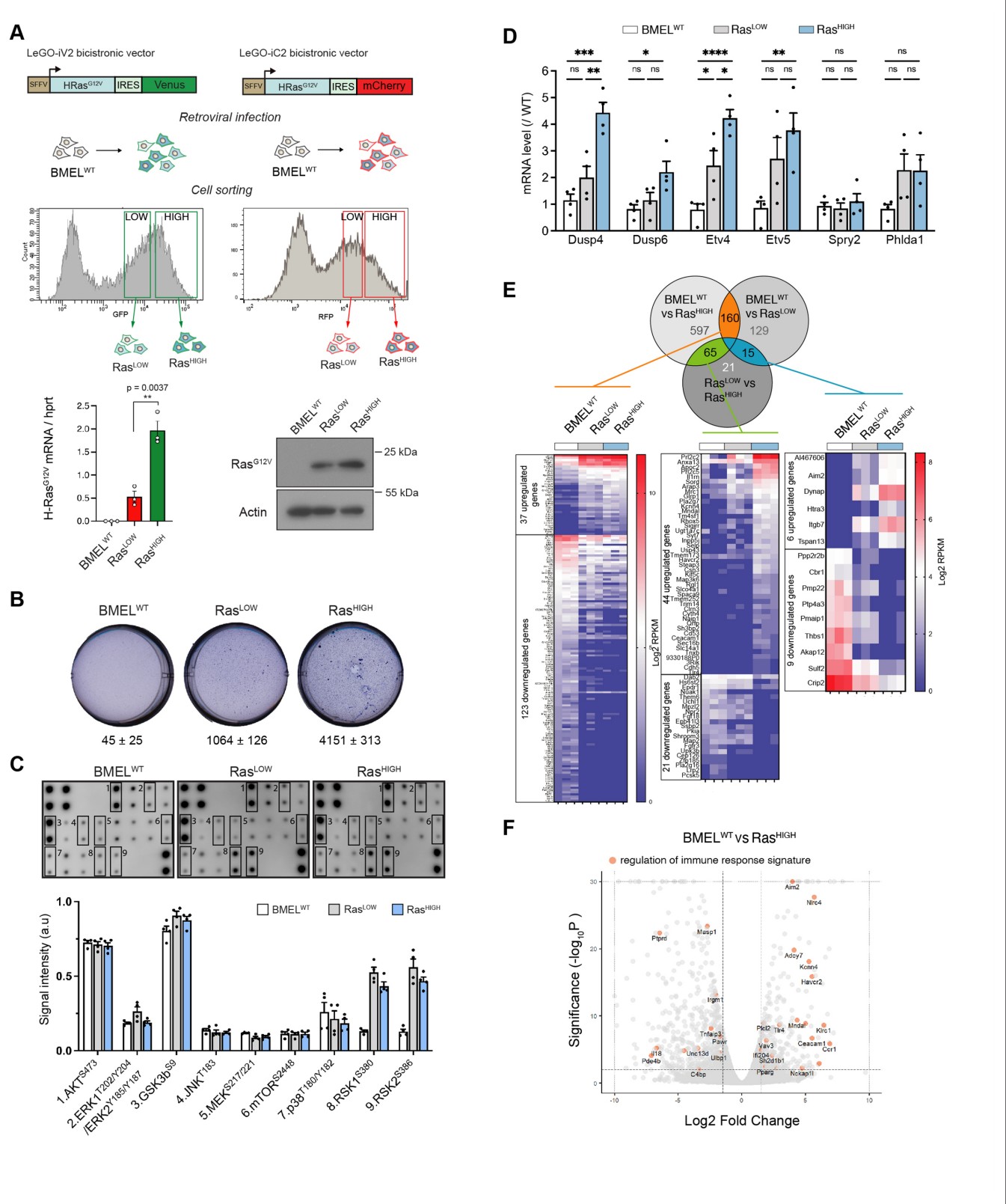

**Figure 1.** Ras[G12V] oncogenic dosage controls the phenotype of bipotent mouse embryonic liver (BMEL) cells. (**A**) Bicistronic vectors were used to transduce BMEL cells that were sorted by flow cytometry according to fluorescence intensities, giving rise to Ras[LOW] and Ras[HIGH] populations labelled with either Venus or mCherry. Ras[G12V], Venus and mCherry expression were quantified by RTqPCR (see also *Figure 1—figure supplement 1*), and results for cell lines BMEL[WT], Ras[LOW-Cherry], Ras[HIGH-Venus] are shown as mean ± SEM. p-Value of unpaired t-test statistical test is shown. Differences in Ras[G12V]

*Figure 1 continued on next page*

*Figure 1 continued*

expression were confirmed by western blot analysis (Ras$^{LOW-cherry}$ versus Ras$^{HIGH-Venus}$). Actin was used as loading control. (**B**) Parental non-transformed BMEL ('BMEL$^{WT}$'), Ras$^{LOW}$ and Ras$^{HIGH}$ cells were seeded in soft agar, stained with crystal violet and counted after 21 days of culture. Representative wells and mean colony numbers ± SEM from two independent experiments performed in triplicate are shown. (**C**) Phosphoprotein array performed on BMEL$^{WT}$, Ras$^{LOW}$, and Ras$^{HIGH}$ cell lines. Bottom panel shows the quantification of duplicates from two independent experiments. a.u.: arbitrary units. (**D**) qPCR quantification of Erk target genes signature in Ras$^{LOW}$ and Ras$^{HIGH}$ BMEL cells normalized to expression in parental BMEL$^{WT}$ cells. Mean and SEM from three to four independent experiments are shown. Statistical significance from unpaired t-test are indicated. (**E**) Venn diagram and heatmaps showing log2 RPKM values of genes regulated by Ras$^{G12V}$ in BMEL cells detected by an RNAseq transcriptomic analysis (log2FC >1.5, p-value <0.01). Genes expressed at low levels (all RPKM log2 values <1) were removed from the analysis. From left to right: 160 genes were modulated by Ras similarly in Ras$^{LOW}$ and Ras$^{HIGH}$ populations, 65 genes were altered only in the Ras$^{HIGH}$ cells and the expression of 15 genes was gradually modulated in the Ras$^{LOW}$ and Ras$^{HIGH}$ populations. (**F**) Volcano plot presentation of deregulated genes in Ras$^{HIGH}$ cells versus parental BMEL cells. Genes belonging to regulation of immune response signature (GO:0050776) are highlighted in orange. ns: not significant, *<0.05, **<0.01, ***<0.001, ****<0.0001.

The online version of this article includes the following source data and figure supplement(s) for figure 1:

**Source data 1.** Raw western blots for RasG12V expression and phosphoproteome arrays.

**Figure supplement 1.** Bipotent mouse embryonic liver (BMEL) clones do not express the p16$^{INK4A}$ tumour suppressor.

**Figure supplement 2.** Transgenes expression levels in sorted cell populations.

**Figure supplement 3.** Ras$^{G12V}$ expression level does not change bipotent mouse embryonic liver (BMEL) proliferation rates ex vivo.

indicating that high level of Ras signalling conferred a selective advantage in vivo. To validate this observation, we next performed orthotopic injections of a 1:1 mix of Ras$^{HIGH}$-Venus:Ras$^{LOW}$-mCherry cells. The in vivo seeding efficiencies of Ras$^{HIGH}$ and Ras$^{LOW}$ cells were indistinguishable, since equal numbers of Venus and mCherry-labelled cells were detected at early times (5 days) post-injection (*Figure 2B*). However, as the tumours grew, a clear imbalance between the two populations became apparent, with Ras$^{HIGH}$ cells becoming the dominant population in tumours already at day 10, which was further confirmed in full-grown tumours 21 days post-injection (*Figure 2B*). This was not due to an artefactual effect of the fluorescent markers, as switching the fluorescent labels between the Ras$^{HIGH}$ and Ras$^{LOW}$ cells had no impact on the dominance of Ras$^{HIGH}$ cells. Furthermore, cells labelled with Venus or mCherry that expressed comparable Ras levels contributed equally to orthotopic tumours (*Figure 2B* lower).

We next enquired about the selective processes within each (i.e. Ras$^{HIGH}$ and Ras$^{LOW}$) subpopulation within the tumours. Of note, and as expected from the gating in the cell sorting (*Figure 1A*), while the mean Ras$^{G12V}$ level was significantly different, there was an overlap of the oncogene dosage between the two injected populations (*Figure 1—figure supplement 2*). Tumours collected at day 21 post-engraftment were dissociated, and Venus+ or mCherry+ cells were sorted by flow cytometry (*Figure 2C* left panel). Interestingly, RTqPCR analysis specific for the Ras$^{G12V}$ revealed that both cellular populations composing the tumour expressed identical level of the oncogene (*Figure 2C* right panel). This suggests that selective pressures operating during the in vivo tumour growth allow the expansion of a population with a defined level of Ras oncogene expression, independently of which parental population they originate from. Since fewer cells in the Ras$^{LOW}$ population fell within the range of this 'optimal' oncogene dosage, they were underrepresented in the fully grown tumour at the end of the experiment. Finally, cell lines established from the orthotopic tumours maintained the high level of Ras expression in vitro (*Figure 2D*), providing further support to the idea of pre-existing clones undergoing selection in vivo.

## Site specific tumour microenvironment selects distinct levels of oncogene dosage

Human HCC give rise mainly to intrahepatic metastases, but extrahepatic invasion also occurs (*Katyal et al., 2000*). This is also the case in our animal model of HCC orthotopic allografts. Indeed, in addition to intrahepatic tumour spread, we consistently observed tumours in the peritoneal cavity, morphologically resembling the paired liver tumours (*Figure 3A*). Extrahepatic tumours likely arose from cell leakage during the surgical procedure of intrahepatic injection, since peritoneal lavage performed after surgery contained low but detectable numbers (<0.5%) of Venus-labelled cells. In this scenario, tumour initiation would be expected to be synchronous at the two locations.

To gain further insight into the characteristics of tumour growth in the primary and metastatic sites, we analysed hepatic and peritoneal tumours generated following injections of either Ras$^{HIGH}$ or

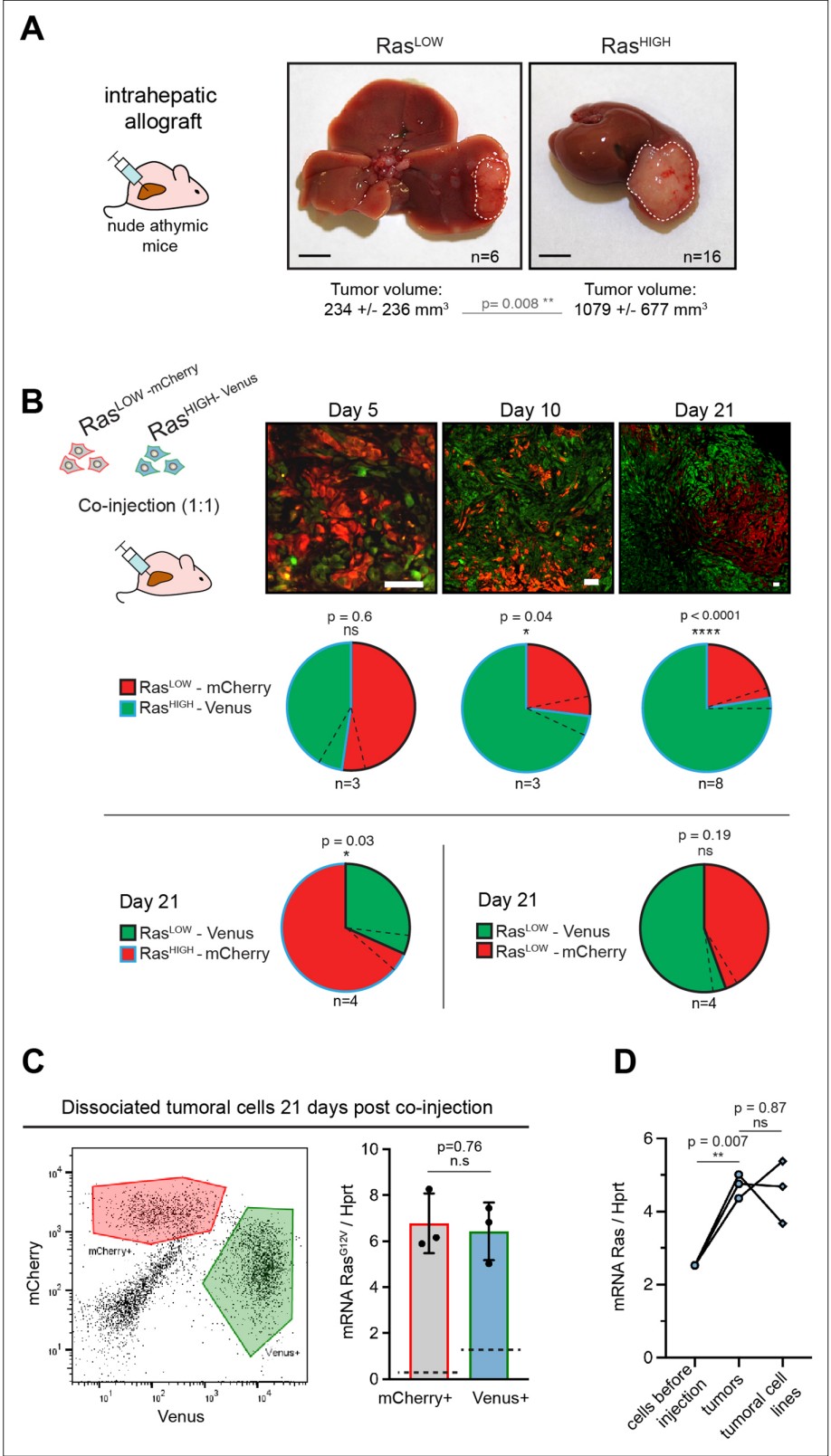

**Figure 2.** High level of Ras expression confers a selective advantage during in vivo tumour growth. (**A**) Macroscopic liver tumours 3 weeks after intrahepatic injection of $10^5$ Ras^LOW or Ras^HIGH bipotent mouse embryonic liver (BMEL) cells, as indicated. Mean ± SD volumes are indicated, with statistical significance of unpaired t-test. Scale bars: 5 mm. (**B**) Mice were injected orthotopically with a 1:1 mix of Ras^HIGH-Venus (green) and Ras^LOW-mCherry (red)

*Figure 2 continued on next page*

*Figure 2 continued*

cells and sacrificed at day 5, 10, and 21, as indicated. Representative images of tumour sections are shown in the upper panels. Different magnifications were used to take into account the tumour growth (scale bars: 50 μm). Contribution of Ras$^{HIGH}$ and Ras$^{LOW}$ cells to the tumour composition was estimated by image analysis and flow cytometry, which gave identical results. Mean ± SEM (dashed lines) are shown. The significance of deviations from the theoretical value of 50% (i.e. no selective advantage) was calculated by the one-sample t-test. (**C**) Cells from dissociated tumours were sorted by flow cytometry (left panel) and the mean expression of Ras$^{G12V}$ in Venus- and mCherry-labelled populations was quantified by RTqPCR. Mean ± SEM from three independent tumours are shown. Dashed lines indicate the mean level of expression of Ras$^{G12V}$ in the population of cells prior to injection. (**D**) Mean Ras$^{G12V}$ expression level quantified by RTqPCR in Ras$^{HIGH}$ cells before injection, cells freshly isolated from the tumours as well as in cell lines isolated from tumours kept in culture for 14 days. Unpaired t-test statistical significance is indicated. ns: not significant, *<0.05, **<0.01, ****<0.0001.

---

Ras$^{LOW}$ cell populations. Strikingly, whereas Ras$^{LOW}$ cells gave significantly smaller hepatic tumours than Ras$^{HIGH}$ cells, this difference was abolished with regard to peritoneal tumours (*Figure 3B*). Altogether, these results confirm that the Ras$^{LOW}$ population contained fewer cells capable of initiating and/or sustaining tumour growth in the liver. In contrast, there appears to be no selective advantage for Ras$^{HIGH}$ cells to establish tumours in the peritoneal cavity.

To further investigate the hypothesis that tumourigenesis at both tissue locations was regulated by the intensity of Ras signalling, we improved the precision of the quantification of the in vivo oncogenic dosage. To do so, we used flow cytometry to separate tumoural from stromal cells derived from liver primary tumours and from the matched peritoneal ones. RTqPCR analyses confirmed the increased oncogenic dosage in the in vivo-grown cells compared to the in vitro-grown parental ones and revealed a significantly higher mean level of Ras$^{G12V}$ expression in the primary versus the peritoneal tumours (*Figure 3C*). Strikingly, the mean level of the oncogene expression was again independent of the origin of the engrafted cells (i.e. Ras$^{HIGH}$ versus Ras$^{LOW}$ populations). Overall, these data support the conclusion that tissue-specific selective pressures favour distinct oncogenic Ras levels.

To address the mechanistic basis for this phenomenon, we performed transcriptomic profiling of Venus+ tumour cells isolated from the primary and the metastatic sites. This analysis identified approximately 160 genes differentially expressed in the liver and peritoneal tumour cells (log2 fold change >1; p<0.05; *Supplementary file 1d*). GO analysis using GSEA identified 27 enriched gene sets (25 in liver and 2 in peritoneum; p-value <0.01 and FDR <0.1. *Supplementary file 1e*). Importantly, it confirmed the increase of MAPK signalling in the liver and identified apoptotic processes and the regulation of the immune response among the gene sets differentially regulated in the two locations (*Figure 3D*). As expected, comparison of transcriptomic profiles from stromal cells of liver and peritoneal tumours pinpointed higher diversity (2260 genes for log2 fold change >1; p<0.05; *Figure 3D*).

Both the hepatic and the peritoneal tumours were highly proliferative (*Figure 3E*). Interestingly, while the cultured BMEL cells expressing either high or low levels of the Ras$^{G12V}$ oncogene have comparable, low, apoptotic indexes, hepatic tumours contain a higher proportion of cleaved caspase 3-positive cells than their peritoneal counterparts (*Figure 3E*). Moreover, in the liver, and much less so in the peritoneum, caspase 3-positive cells often surround large necrotic areas, indicating that the cell death in an expanding liver tumour is quite substantial, suggesting an evolutionary trade-off for the high oncogene dosage in the liver tumours.

## Immune cell contexture differs in primary and metastatic-like tumours

We next interrogated the interplay between the tumour cells and their local microenvironments that shape the oncogenic expression profile. The transcriptomic profiling of the stromal component of liver and peritoneal tumours revealed a number of differentially expressed genes, as expected for distinct tumour locations. Cibersortx deconvolution of the immune component of the stroma (*Newman et al., 2019*) did not show strong differences in the numbers of intra-tumoural macrophages at the two locations (*Figure 4A*). However, independent RTqPCR profiling of macrophage polarization markers did suggest a more immunosuppressive, less inflammatory microenvironment in the peritoneum (*Figure 4B*, *Figure 4—figure supplement 1*).

Interestingly, the presence of immunosuppressive tumour-associated macrophages (TAMs) did not deter from other antigen-presenting cells activity in the peritoneal tumour microenvironment (TME). Indeed, our data suggested that a larger proportion of mature, activated dendritic cells (DCs), as

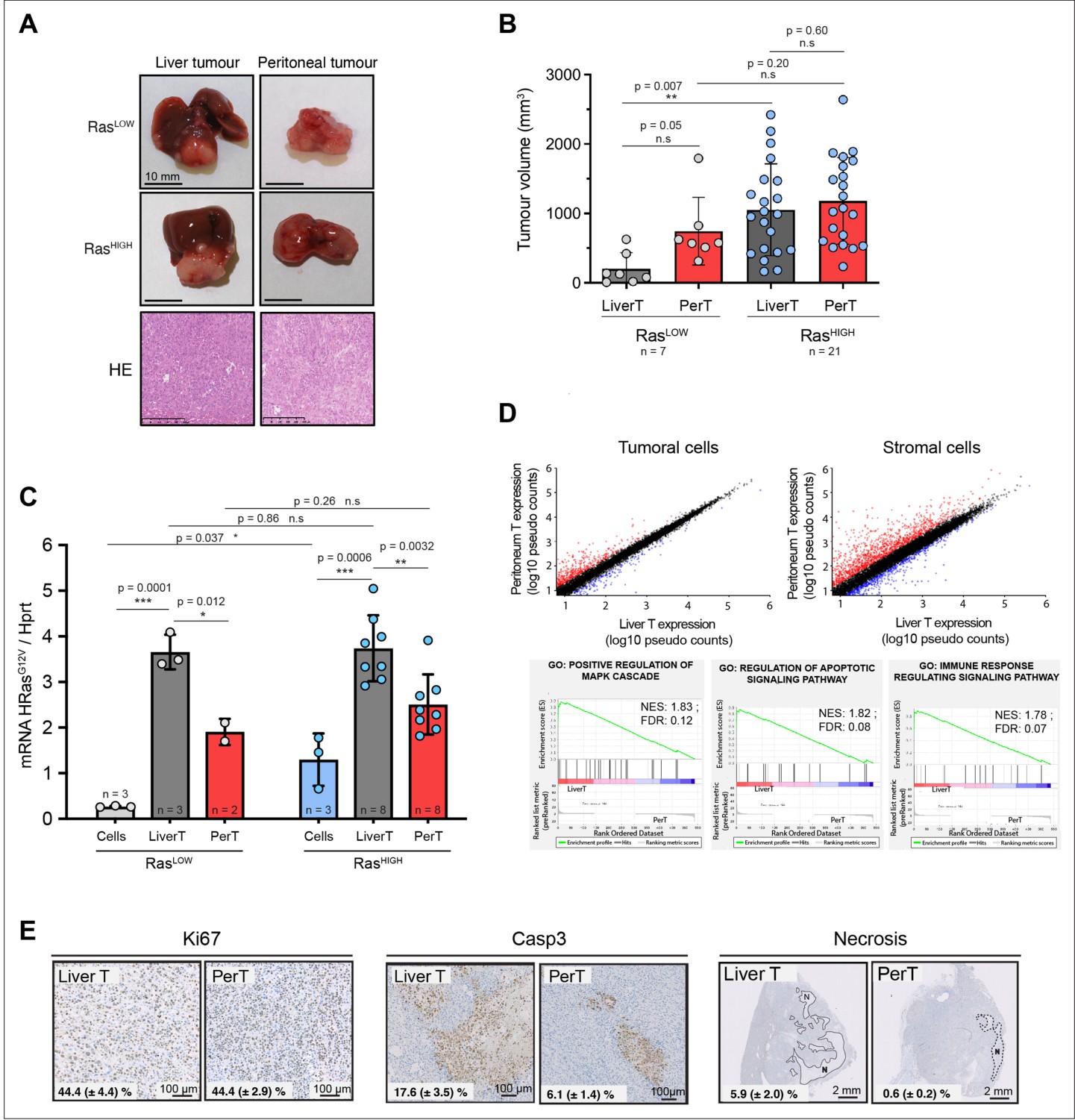

**Figure 3.** Distinct tumour environments select a site-specific range of optimal Ras signalling intensity. (**A**) Representative images of paired liver and peritoneal tumours collected 3 weeks after injection of BMEL-Ras$^{LOW-Cherry}$ and Ras$^{HIGH-Venus}$ cells, as indicated. Lower panel shows hematoxylin/eosin (HE) stainings. (**B**) Quantification of liver and peritoneal tumour volumes from three independent experiments (mean ± SD). LiverT: liver tumour, PerT: peritoneal tumour. Unpaired t-test p-values are indicated. (**C**) Liver and peritoneal tumours resulting from injection of Ras$^{LOW-mCherry}$ or Ras$^{HIGH-Venus}$ cells, as indicated, were dissociated and fluorescent cells were sorted by flow cytometry. Ras$^{G12V}$ expression levels in cells prior to injection and in cells purified from the tumours were quantified by Taqman RTqPCR. Mean ± SD are shown. Unpaired t-test p-values are shown. (**D**) Scatter plots representation of gene expression in tumoural and stromal cells from liver and peritoneal tumours. Pseudo counts mean value from four independent samples is shown. Genes upregulated or downregulated (ratio >2) in perT versus LiverT are shown in red or blue, respectively. Lower panels show the enrichment plot

*Figure 3 continued on next page*

*Figure 3 continued*

for the indicated gene ontology (GO) terms: positive regulation of MAPK cascade, regulation of apoptotic signalling, immune response regulation. (**E**) Immunohistochemical analysis of cell proliferation, apoptotic cell death, and necrotic areas in Ras[HIGH] liver and peritoneal tumours. Numbers on each panel represent mean ± SEM from the analysis of duplicate tumour sections from seven animals. ns: not significant, *<0.05, **<0.01, ***<0.001.

compared to immature ones, were present in this organ (*Figure 4A*). Flow cytometry analyses confirmed that while the total number of DCs (defined as CD45+ CD11b+ CD11c+ MHCII[high] CD24+F4/80-) was not significantly different between the two TME, these cells showed an increase in proliferation in the peritoneum (*Figure 4C*). Furthermore, the peritoneal DCs contained lower levels of the milk-fact globule-EGFVIII (MFG-E8), a factor known to limit DC immunogenic potential (*Figure 4D*; *Baghdadi et al., 2012*; *Peng and Elkon, 2012*).

The phenotypes of macrophages in matched liver and peritoneal tumours also showed some interesting differences. The content of peripherally recruited monocyte-derived macrophages presenting low expression of F4/80, defined as CD45+CD11b+Ly6G-Ly6C-F4/80[low], was significantly enriched in liver tumours (*Figure 4E*). On the other hand, the differentiated, tissue resident-like, F4/80[high] macrophages showed a tendency for enrichment in the peritoneal HCC. These results suggest that macrophage subsets are distinct in the liver and peritoneal TME, with tissue-resident, F4/80[high] cells, which are part of organ homeostasis maintenance (*Dou et al., 2019*) dominating the peritoneal TME while abundant F4/80[low] immature TAMs infiltrate the liver. Interestingly, F4/80[high] peritoneal macrophages displayed a superior proliferative capacity (Ki67+) and activation phenotype (MHCII+, CD115+) compared to liver macrophages (*Figure 4F*). Coherent with our findings that more activated and less immature DCs are present in peritoneal TME (*Figure 4A and D*), these results suggest that the liver TME may represent a more immune tolerant environment for Ras[HIGH] tumour cell outgrowth compared to the peritoneum, in which the activation levels of the Ras oncogene do not affect the dynamics of tumourigenesis. Thus, through both flow cytometry and RNAseq deconvolution analyses, our results highlight the differences in stromal composition of the tumours in the two locations, showing the complex interplay of innate immune cells that may participate in shaping the clonal selection in the tumour.

We reasoned that a likely consequence of increased numbers of activated DCs would be an improved efficiency of antigen presentation, thereby allowing a more efficient tumour clearance in the peritoneum.While such phenotype is not relevant in immunodeficient mice that are unable to mount a T-cell response, it should be detectable in immunocompetent animals. In order to test this prediction, we compared the outcomes of orthotopic injections of tumour cells in immunocompetent C57BL/6 and immunodeficient nude mice. For these autologous allografts, we took advantage of a C57BL/6 primary cell line that we derived from a tumour established by a hydrodynamic gene transfer of constitutively active Ras and a simultaneous CrispR/Cas9-mediated inactivation of the p53 tumour suppressor (*Bacevic et al., 2019*). Orthotopic allografts of 5000 cells performed in parallel on immunocompetent and immunodeficient mice gave rise to comparable macroscopic liver tumours within 3 weeks (*Figure 4G*). However, in contrast to nude mice, of which 50% developed macroscopic peritoneal tumours, no growth at a metastatic location was detected in immunocompetent recipients. These results are consistent with the hypothesis that a more effective antigen presentation in the peritoneum limits metastatic tumour growth in immunocompetent animals.

## Ras signalling modulates tumour cells interactions with stromal cells

We next questioned the mechanistic bases of the tumour and stroma crosstalk that might shape the quantitative differences in the oncogenic pathway activation. To do so, we looked for genes whose expression is specifically induced by high-intensity Ras signalling, both in the parental BMEL-Ras[HIGH] cells and in the liver, as compared, respectively, to the Ras[LOW] cells and the peritoneal tumours (*Figure 5A*).Out of four genes (*Ceacam1, Csn3, Selp, Tmem252*) that corresponded to this criterion, we focused our attention on CEACAM1 (carcinoembryonic antigen-related cell adhesion molecule 1), a gene widely expressed in many cancer types and whose expression level has been reported to correlate with tumour progression (*Dankner et al., 2017*). It is also expressed on the surface of several immune cell subsets, including macrophages, T lymphocytes, and NK cells, where homophilic interactions with the CEACAM1+ tumour cells abrogate NK-mediated cytotoxicity (*Helfrich and Singer, 2019*; *Markel et al., 2002*). In order to confirm that the observed Ras-mediated transcriptional

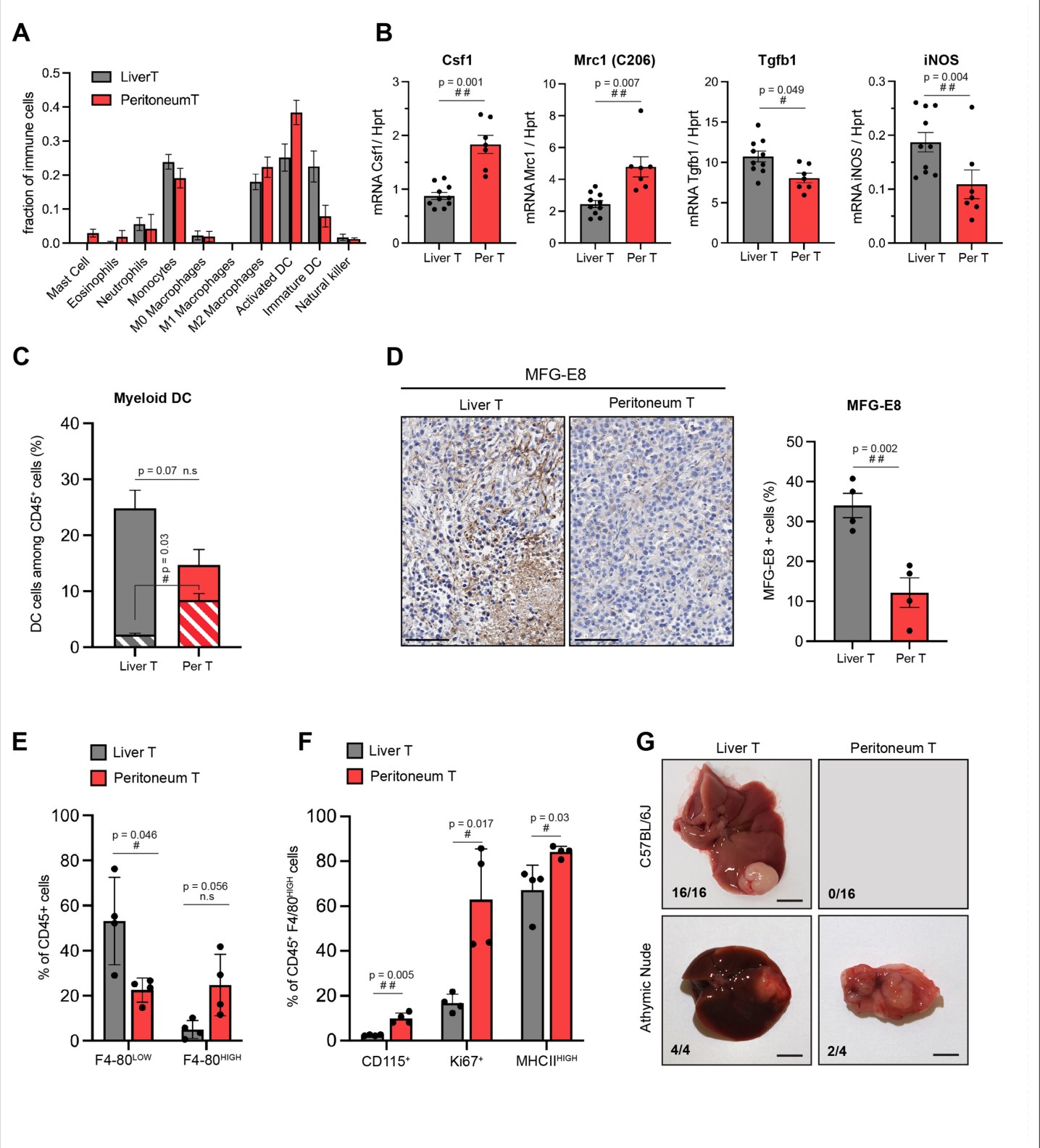

**Figure 4.** Site-specific immune contexts of primary and metastatic tumour locations. (**A**) Cibersortx algorithm was used to perform cell type deconvolution of RNAseq data from stroma of matched liver and peritoneal tumours (n=4). Ten immune cell populations were included in the signature, as indicated. (**B**) Macrophage polarization marker expression was analysed by RTqPCR performed on RNA isolated from stromal cells sorted from liver and peritoneal tumours. (**C**) Dendritic cells (DCs) (CD45+, CD11c+, MHCII$^{high}$, F4/80-, CD24+) from liver and peritoneal tumours were analysed

*Figure 4 continued on next page*

*Figure 4 continued*

by flow cytometry. The histogram represents the percentage of DCs among the total CD45+ population, indicating the proliferating (Ki67+, dashed bars) and non-proliferating (Ki67-, filled bars) DCs. (**D**) Immunohistochemistry performed on liver and peritoneal tumour sections using anti-MFG-E8 antibody, quantification of positive staining is shown in the right panel. Scale bars: 100 μm. (**E**) Flow cytometry analysis of CD45+CD11b+Ly6G-Ly6C-F4/80$^{low}$ (monocyte-derived macrophages) and CD45+CD11b+Ly6G-Ly6C- F4/80$^{high}$ (tissue-resident macrophages) from liver and peritoneal tumours. (**F**) Quantification of CD115+, Ki67+, and MHCII$^{high}$ cells among CD45+CD11b+Ly6G-Ly6C-F4/80$^{high}$ macrophages. (**G**) 5000 cells (NRAS$^{G12D}$ p53$^{KO}$) were injected orthotopically in C57BL/6J mice and athymic nude mice, as indicated. Macroscopic images of representative liver and peritoneal tumours collected 3 weeks after injections (scale bar = 5 mm). Numbers indicate the fraction of inoculated animals that developed tumours. All quantitative analyses were subjected to a paired Student's t-test. p-Values: #<0.05, ##<0.01, ###<0.001.

The online version of this article includes the following figure supplement(s) for figure 4:

**Figure supplement 1.** Myeloid infiltrates differ in the microenvironments of hepatic and peritoneal tumours.

activation translates into CEACAM1 expression on the surface of cancer cells, we performed FACS analyses on cells dissociated from the liver and peritoneal tumours. As expected, the majority of the non-hematopoietic (CD45$^{neg}$) cells in both tumoural locations expressed the Venus fluorescent protein and were therefore the Ras$^{G12V}$ expressing tumoural cells. CEACAM1 expressing cells were highly enriched in this population in the liver as compared to peritoneal tumours, corresponding respectively to 30.1±2.4% (mean ± SEM) and 14.2±3% (mean ± SEM) of CD45$^{neg}$ cells (*Figure 5B*). Strikingly, the mean fluorescent intensities of the CEACAM1 labelling in the Venus+ populations were indistinguishable between the tumours at the two locations. This is consistent with the transcriptomic data indicating that high level of Ras signalling is needed to activate CEACAM1 gene expression in BMEL cells (*Figure 1E and F*) and confirms that the cells that have the required Ras dosage are more abundant in the liver than in the peritoneal tumours. Interestingly, while difference in the NK cell content in the primary and peritoneal tumours did not reach statistical significance in the deconvolution of stromal cells RNAseq (*Figure 4A*), immunofluorescence staining revealed distinct distribution of these cells: whereas NK infiltrate the hepatic tumours, they apparently tend to remain in the periphery of the peritoneal ones (*Figure 5—figure supplement 1*).

We reasoned that if the high Ras oncogenic dosage was instrumental in resisting NK-cell mediated cytotoxicity in the liver tumours, the different sensitivities to NK exposure should also be detected ex vivo for the Ras$^{HIGH}$ versus Ras$^{LOW}$ cells. We have therefore isolated clones expressing defined levels of Ras, representative of high and low populations (3.20±0.62 and 0.50±0.14 Ras/Hprt mRNA levels, respectively). NK cells were isolated from spleen of athymic nude mice and used in ex vivo cytotoxicity assays by co-culture with clones of Ras$^{HIGH-Venus}$ and Ras$^{LOW-mCherry}$ cells (*Figure 5C*). As predicted by our hypothesis, the Ras$^{HIGH}$ cells were indeed less sensitive to NK-mediated cell death than their Ras$^{LOW}$ counterparts.

To further investigate the involvement of NK cells in the selection of Ras$^{HIGH}$/CEACAM$^{HIGH}$ cells in liver tumours, we next analysed tumourigenesis in NOD scid gamma (NSG) mice. These animals are deeply immunosuppressed: in addition to lacking mature B and T cells they are notably devoid of NK cells (*Yu et al., 2008*). The tumour volumes were comparable in the NSG and in the nude mice. As in the nude mice, the NSG background was permissive for peritoneal metastatic-like growth, however, the majority of the animals displayed comparable tumour volumes at the two locations (*Figure 5D*). Consistently, the number of caspase 3-positive cells was low for both liver and peritoneal tumours in NSG animals (*Figure 5E*). These results are consistent with the idea that the discrimination against the Ras$^{LOW}$ cells, both in the peritoneum and in the liver, is alleviated in the NSG background. In support of this contention, compared to the nude mice, the hepatic tumours in the NSG background tolerated lower levels of CEACAM1 expression, while there was no significant difference for the peritoneal tumours between the two genetic backgrounds. Of note, however, the difference between CEACAM1 expression in the hepatic and peritoneal tumours was diminished, but not abolished in the NSG animals (*Figure 5F*). Altogether, these results suggest that CEACAM1-driven NK inhibition participates in the clonal selection, but is not the sole element of the advantage afforded by the high oncogene dosage.

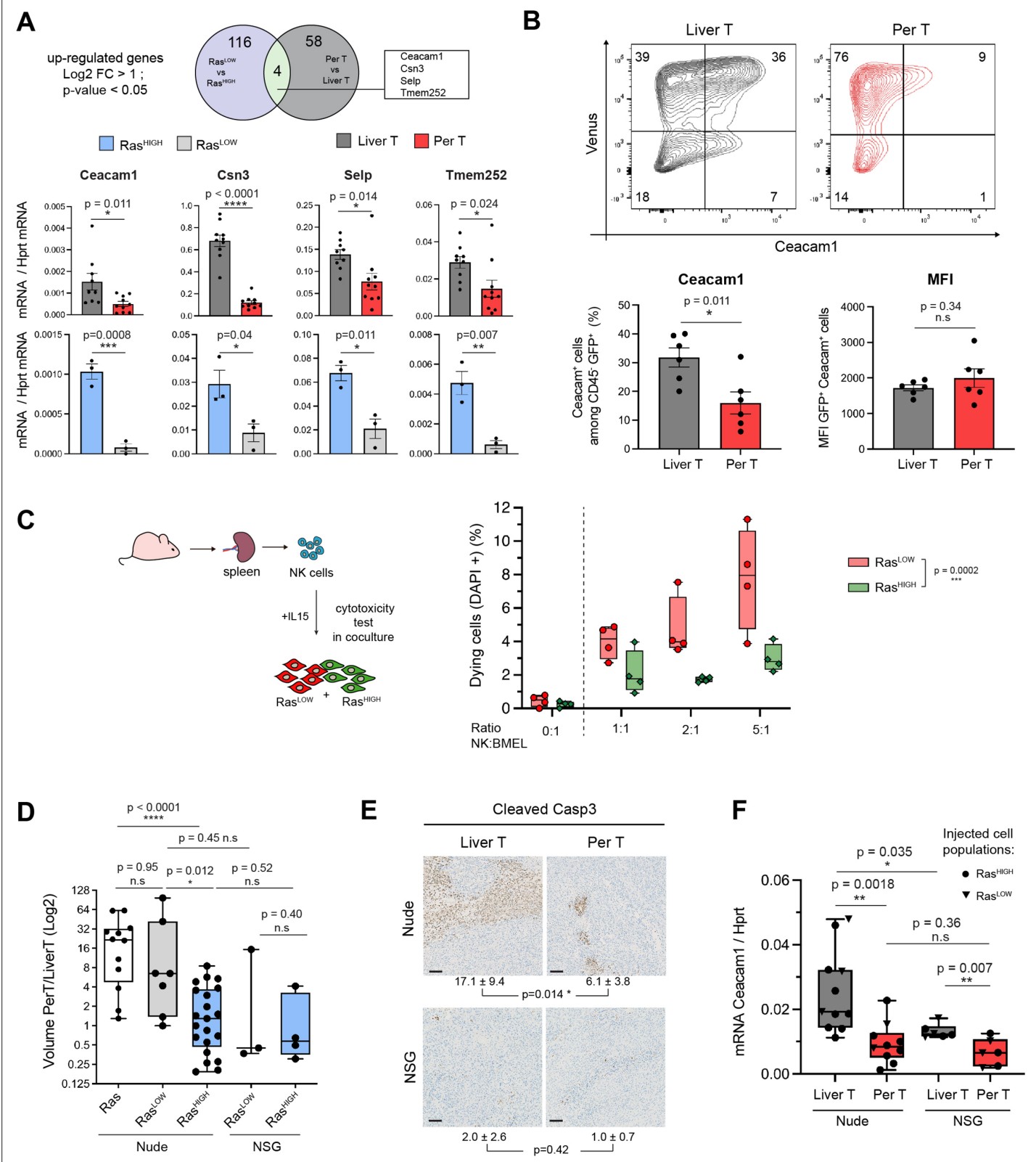

**Figure 5.** Natural killer (NK) cells contribute to clonal selection in liver tumours. (**A**) Venn diagram representing differentially expressed genes (log2FC >1, p-value <0.05) from RNAseq data of Ras^LOW versus Ras^HIGH bipotent mouse embryonic liver (BMEL) cells and sorted tumoural cells derived from peritoneal versus liver tumours. Bottom panel: RTqPCR quantification of the four genes upregulated both in the hepatic versus peritoneal tumours and in RasHIGH versus RasLOW cells. Upper panels: analysis of FACS-sorted cells isolated from tumours, lower panels: analysis of ex vivo grown

*Figure 5 continued on next page*

Figure 5 continued

cells. (B) Flow cytometry analysis of tumoural cells (CD45- Venus+ Ceacam1+) isolated from the liver and the peritoneal tumours. Numbers indicate percentage of cells in each quadrant. Bottom panels show quantification (left) and mean fluorescent intensity (MFI, right) of the Ceacam1 expressing tumour cells. (C) NK cytotoxicity test: NK cells isolated from the spleen of nude mice were activated with IL15 and incubated with co-cultures of Ras$^{HIGH}$: Ras$^{LOW}$ clones (1:1). Synopsis of the experiment (left panel) and FACS quantification of DAPI-positive cells (right panel). Values are duplicates from two independent experiments. p-Value from two-way ANOVA test is indicated. (D) Ratios of tumour volumes from matched peritoneal and liver tumours (PerT/LiverT) originating from non-sorted BMEL-Ras cells, Ras$^{LOW}$ or Ras$^{HIGH}$ populations, as indicated, orthotopically inoculated into athymic nude or NOD-SCID-gamma (NSG) mice. (E) Immunohistochemical analysis of activated caspase 3 performed on liver and peritoneal tumours in athymic nude and NSG mice. Mean ± SD values of positive cells from five animals are indicated. Scale bar: 100 µm. (F) RTqPCR quantification of Ceacam1 expression in sorted tumour cells from athymic nude and NSG mice. Unpaired Student's t-test, p-values. ns: not significant, *<0.05, **<0.01, ***<0.001, and ****<0.0001 p-values.

The online version of this article includes the following figure supplement(s) for figure 5:

**Figure supplement 1.** Differences in natural killer (NK) cells infiltration between liver and peritoneal tumours.

## Discussion

Multiple environmental and cell-autonomous mechanisms, organized in negative and positive feedback loops, cooperate to allow an exquisite regulation of the intensity and duration of signalling, the variation of which can give rise to specific and distinct cell fates. The Ras/MAPK Erk pathway is a prototypic example of such complex regulatory circuits (*Murphy and Blenis, 2006*), whose output is also relevant in the context of tumourigenesis. This has been shown for example in response to the loss of anchorage, where strong Erk signal is protective, while a moderate one is required for anoïkis (*Zugasti et al., 2001*). However, excessive Erk activation can be detrimental to the cell, provoking cell cycle arrest or senescence (*Serrano et al., 1997*; *Woods et al., 1997*). More recent single-cell functional analyses further support the evidence for cell-autonomous sensitivity to signal intensity, but also reveal the heterogeneity of cellular response, both in vitro and in vivo (*Gross et al., 2019*; *Kumagai et al., 2015*; *Molina-Sánchez et al., 2020*).

Our understanding of how the availability of extracellular stimuli, or the deregulation of their sensors, translate into cellular responses is still incomplete. For example, although only 5% of the small GTPase Ras needs to be activated to give rise to full activation of Erk (*Hallberg et al., 1994*), a clear dosage effect has been reported in pancreatic and lung tumours for activating Ras mutations (*East et al., 2021*; *Kerr et al., 2016*; *Mueller et al., 2018*). Moreover, amplifications of the wild-type allele of Ras that give rise to an allelic imbalance and the consequent modulation of the activity of the oncogenic Ras mutants result in distinct cellular phenotypes (*Ambrogio et al., 2018*; *Bremner and Balmain, 1990*). This further argues for the physiological importance of precise Ras dosage in a growing tumour.

Of note, whereas activating Ras mutations are rare in HCC (*Zucman-Rossi et al., 2015*), alterations in the intensity of MAPK Erk signalling participate in shaping the tumoural phenotype, as revealed by inactivating mutations of Rsk2, which is both an ERK target and a negative regulator of the pathway (*Chan et al., 2021*). Moreover, a recent analysis of transcriptional signatures classified HCC among tumours with significant Ras pathway activation phenotype (*East et al., 2021*). Because the same report highlighted the predictive value of strong versus weak Ras-driven transcriptional signature, it is relevant to model these phenotypes in HCC by controlled dosage of the Ras oncogene, as we have done in this study.

Our exploration of signal intensity that is optimal for tumour growth in the context of its specific microenvironment led to a description of a novel feature of tumour heterogeneity, manifested by quantitative dosage differences of oncogenic signalling in tumours developing at primary and metastatic locations. We note that the peritoneal tumours in our animal model likely do not result from a bona fide metastatic process. Nevertheless, they do represent tumoural growth at an extrahepatic site and are thus a convenient model to analyse primary and metastatic-like tumour growth within the same animal.

Although several oncogenic signalling pathways lie downstream of Ras (*Sanchez-Vega et al., 2018*), in our experimental model Ras$^{G12V}$ dosage correlates with the MAP kinase Erk activity, but not with the phosphoinositide 3-kinase. Whether such uncoupling is a general feature of HCC or simply a reflection of a strong PI(3)K/Akt signalling present in the non-transformed BMEL parental cells remains to be established.

Comparing tumours formed by BMEL-Ras cells expressing a wide range of Ras[G12V] oncogene ('parental Ras' cells), with those initiated by cells with the narrower range of signalling intensities (Ras[HIGH] and Ras[LOW] cells) suggests the existence of a trade-off in the tumours' evolutionary history. Indeed, we have shown that cells expressing high oncogene levels are globally favoured in the primary liver tumours, however, their selective advantage is curtailed by a higher probability of undergoing programmed cell death. While the propensity to activate the senescence and/or apoptosis in response to oncogenic Ras signalling is well known (*Serrano et al., 1997*; *Woods et al., 1997*), the observation of diminished cell death in the tumours growing in NSG animals points to an additional, non-cell-autonomous, mechanism that drives this response.

The immune response to cancer comprises a complex interplay of innate and adaptive components of immunity (*Fridman et al., 2012*; *Molina-Sánchez et al., 2020*; *Nguyen et al., 2021*). Concentrating our study on animals devoid of an adaptive immune response gave us access to interactions between the tumour and its innate immune component. Our results point to key differences in myeloid cell population subsets in the liver and peritoneum of tumour-bearing mice. Notably, the content of tumour-infiltrating, immunosuppressive macrophages, whose function in promoting liver carcinogenesis is well recognized (*Dou et al., 2019*; *Ringelhan et al., 2018*), was enhanced in Ras[HIGH] liver tumours, and correlated with an increased fraction of immature DCs when compared to the peritoneal TME. These results suggest that rather than total numbers of each myeloid populations, it is the phenotype of these cells, driven by the local cues, that regulates their ability to further engage an anti-tumour immunity and control tumour outgrowth. In line with this observation, recent studies have shown that identical oncogenic drivers have strikingly different consequences on the content and activation of TME cells in lung or pancreatic adenocarcinomas (*Kortlever et al., 2017*; *Sodir et al., 2020*). Hence, in addition to the tumour cells' genetic landscape itself (*Wellenstein and de Visser, 2018*), the downstream activation of signalling nodes pertaining to dosage of a specific oncogene should be considered as a tumour-intrinsic feature shaping the TME, as recently also reported for pancreatic cancer (*Ischenko et al., 2021*).

It is of particular interest to consider the role played by NK cells as part of the organismal defence against HCC. NK activities are strongly dependent on, as yet incompletely understood, organ-specific features (*Shi et al., 2011*). Resident NK cells acquire cytolytic functions mediated through TNFα and granzyme B during hepatotropic viral infections, cirrhotic pathology, and HCC (*Sällberg and Pasetto, 2020*). The viral adaptation to the NK-mediated attack includes the suppression of their function through the CEACAM1-mediated blockade of their activity (*Suda et al., 2018*). Our results are fully consistent with the idea that a similar mechanism of immune escape operates in primary tumours of HCC. In contrast to the liver, the peritoneal resident cells resemble immature splenic NK cells (*Gonzaga et al., 2011*), which may account for a lesser involvement of these cells in anti-tumour immunity in this location, as suggested by our data.

How can the organ-specific differences in primary and metastatic tumour growth be exploited in designing novel therapeutic approaches remains an open question. The observation that only a narrow range signalling intensity through a signal transduction pathway is compatible with tumour development raises hopes for the therapeutic efficacy of its modulation, as recently reported for stratification of HCC response to chemotherapeutic intervention based on the inactivating Rsk2 mutations that modulate the strength of the MAPK signalling (*Chan et al., 2021*). Similarly, only lung adenocarcinoma with a strong Ras activating signature respond to inhibitors of the MAPK Erk pathway (*East et al., 2021*).

A key role of NK cells in anti-tumour immunity is today well appreciated (reviewed in *Pende et al., 2019*). Moreover, there is a rising interest in the use of NK cell-based therapies, largely because of their considerably lower toxicity in comparison to T-cell transfer (reviewed in *Xie et al., 2020*), translating into over 200 ongoing clinical trials on solid tumours, including HCC (https://www.clinicaltrials.gov/). Our data suggest that the therapeutic efficacy of NK-based therapy for HCC might be improved by simultaneous inhibition of Erk signalling. Such adjuvant therapy may show clinical benefit, notably for the subset of tumours characterized by strong MAPK signalling.

## Materials and methods

### BMEL preparation and cell culture

BMEL cells were isolated from C57BL/6xC3H E14 embryonic mouse female liver, as described in *Strick-Marchand and Weiss, 2002*. Briefly, livers at 14 dpc were separately dissected, and cell suspension was plated onto collagen-coated 100 mm Petri dishes in hepatocyte attachment medium (Invitrogen, Cergy Pontoise Cedex, France). The next day and thereafter, medium was replaced by RPMI 1640 (Eurobio) containing 10% fetal bovine serum (Eurobio), 50 ng/ml epidermal growth factor, 30 ng/ml insulin-like growth factor II (PeproTech, Rocky Hill, NJ), 10 mg/ml insulin (Sigma), and antibiotics. Clones were picked after 2–3 months of culture and expanded in the same medium on dishes coated with Collagen I (BD Biosciences, Le Pont de Claix, France) in humidified atmosphere with 5% $CO_2$ at 37°C. All cells were tested negative for mycoplasma contamination every week.

### Isolation of cell populations

Human H-Ras[G12V] cDNA sequence was cloned into LeGO-iV2 and LeGO-iC2 bicistronic vectors (Addgene #27344 and #27345, respectively). Retroviruses were produced by JetPEI (Polypus-Transfection) transfection of HEK293T cells with Hras[G12V]-iV2 and Hras[G12V]-iC2 constructs together with Gag/pol and pCAG-Eco packaging vectors. Culture media containing viruses were filtered (0.45 μm) and used to infect BMEL cells. Based on Venus (530/30 nm) and mCherry (610/20 nm) fluorescence intensity, Aria IIU and IIIU cell sorters (Becton Dickinson) were used to isolate Hras[G12V]-LOW and Hras[G12V]-HIGH populations from Hras[G12V]-iV2 and Hras[G12V]-iC2 BMEL cells.

### Western blotting

Protein lysates of cells were prepared with lysis buffer (150 mM NaCl, 50 mM Tris pH 7.5, 1% Triton, 1% SDS; freshly added protease inhibitors cocktail [Roche]). Protein concentrations were determined by BCA protein assay (Pierce Biotechnology). Five μg of proteins were separated by SDS-PAGE (Precast 4–12% gel). Proteins were transferred on nitrocellulose membrane, and loading checked by red ponceau staining. Primary antibody against Ras[G12V] (Cell Signaling #14412, 1:2000) and Actin (Sigma A1978, 1:20,000) were incubated overnight and 1 hr respectively, and revealed with HRP-coupled secondary antibodies plus chemiluminescent detection following classical procedures.

### Soft agar assay

$10^5$ cells/well were seeded in 0.5% agar diluted in RPMI media. One ml media was added to the agar layer and was changed every 3 days. After 3 weeks, colonies were stained with 0.005% crystal violet 4% PFA. Colonies were counted automatically in ImageJ using the 'Analyze particles' function.

### Orthotopic xenografts

All reported animal procedures were carried out in accordance with the rules of the French Institutional Animal Care and Use Committee and European Community Council (2010/63/EU). Animal studies were approved by Institutional Ethical Committee (Comité d'éthique en expérimentation animale Languedoc-Roussillon (#36)) and by the Ministère de l'Enseignement Supérieur, de la Recherche et de l'Innovation (APAFIS#11196-2018090515538313v2). ARRIVE guidelines were followed. Sample sizes were selected to allow these experiments to be sufficiently powered such that differences between experimental groups will be statistically significant with type I error rates <5% (e.g. p<0.05) and with 80% power to detect a standardized difference. Athymic nude (Crl:NU(NCr)-Foxn1nu), NSG (NOD.Cg-PrkdcSCID Il2rgtm1Wjl/SzJ) or C57BL/6J mice from Charles River were anesthetized using intraperitoneal injections of ketamine/xylazine. A 1.5 cm transversal incision of the skin was performed below the sternum's xyphoid process and followed by a transversal incision of the peritoneum, the tip of the liver left lateral lobe was pulled out of the peritoneal cavity. BMEL cells suspension in PBS-20% Matrigel (Corning) were injected in the liver parenchyma. Stitching of the peritoneum then skin was performed using surgical suture (Monosof 5/0 3/8C 16 mm).

### Analysis of tumour cellular composition

Tumours were fixed in 4% PFA for 4 hr and left overnight in 30% sucrose solution. Fixed tissues were placed in OCT cryogenic matrix and frozen in liquid nitrogen. Sections were cut at –20°C and

**Table 1.** List of primers used in RTqPCR.

| Gene name | Forward primer | Reverse primer |
| --- | --- | --- |
| Venus | AAGGCTACGTCCAGGAG | CGGTTCACCAGGGTGTC |
| mCherry | TCCGAGCGGATGTACCC | GGCCTTGTAGGTGGTCT |
| HRas | GAGGATGCCTTCTACACGTT | GCACACACTTGCAGCTC |
| HRas Taqman primers | GGCATCCCCTACATCGAGA | CTCACGCACCAACGTGTAGA |
| Hprt | GCAGTACAGCCCCAAAATGG | GGTCCTTTTCACCAGCAAGCT |
| Hprt Taqman primers | CCTCCTCAGACCGCTTTTT | AACCTGGTTCATCATCGCTAA |
| Phlda1 | GGGCTACTGCTCATACCGC | AAAAGTGCAATTCCTTCAGCTTG |
| Dusp4 | CGTGCGCTGCAATACCATC | CTCATAGCCACCTTTAAGCAGG |
| Dusp6 | GCGTCGGAAATGGCGATCT | ATGTGTGACGACTCGTACAGC |
| Etv4 | CGGAGGATGAAAGGCGGATAC | TCTTGGAAGTGACTGAGGTCC |
| Etv5 | TCAGTCTGATAACTTGGTGCTTC | GGCTTCCTATCGTAGGCACAA |
| Sprty2 | TCCAAGAGATGCCCTTACCCA | GCAGACCGTGGAGTCTTTCA |
| Ceacam1 | CACAGGACCCTATGTGTGTGAAA | CACTGGCTCAAGGACTGTA |
| Ceacam1 Taqman primers | GGGCTGGCATATTTCCTCTATT | GTTGTCAGAAGGAGCCAGATT |
| Csn3 | CAGTCTGCTGGAGTACCTTATG | AGGATTGGCCACAGTATTTACTAT |
| Selp | CACTGGCTCAAGGACTGTA | TCCAGTAGCCAGGCATCTTA |
| Tmem252 | GAACAGGACTGGGCTGAT | CTCTTCATAAGCTGGAGGGTAAA |
| Arg1 | CATTGGCTTGCGAGACG | CCAGCTTGTCTACTTCAGTCAT |
| IL10 | TAATGCAGGACTTTAAGGGTTACT | CATCCTGAGGGTCTTCAGC |
| TGFb1 | TGCTAATGGTGGACCGC | CATGTTGCTCCACACTTGATTT |
| Mrc1 (CD206) | GGTGGGCAGTCACCATA | GGTTCTCCTGTAGCCCAAG |
| Ltbeta | CCTGCTGCCCACCTCATA | CGACGTGGCAGTAGAGGTAATA |
| IL6 | CTAAGGACCAAGACCATCCAAT | GATATGCTTAGGCATAACGCAC |
| TNFa | TCAGTTCTATGGCCCAGACC | GTCTTTGAGATCCATGCCG |
| IL1beta | TTCCCATTAGACAACTGCACTAC | TATTCTGTCCATTGAGGTGGAGAG |
| IL1-rn | AAGCCTTCAGAATCTGGGATAC | GGATGCCCAAGAACACACT |
| iNOS | GTGGTGACAAGCACATTTGG | GTGGTTGAGTTCTCTAAGCATGA |
| p16 INK4 | GAGCAGCATGGAGTCCG | GGGTACGACCGAAAGAGTT |
| p19 ARF | TTGGTGAAGTTCGTGCGAT | TGGTCCAGGATTCCGGT |
| Csf1 | AGCGACCACCCAGGAGTA | AGTTAGTGCCCAGTGAAGATT |

mounted on superFROST slides with ProlonGold mounting media. Whole slide scans were performed with Zeiss Axioscan at 20× magnification and DAPI (405 nm), GFP (488 nm), and Cy3 (550 nm) fluorescent channels. On average 10 sections from different regions of the tumours were analysed per mouse. Quantification was done by delimiting GFP-positive or Cy3-positive regions and measuring their area on ZEN software. Alternatively, tumours were dissociated with mouse tumour dissociation kit (Miltenyi) and analysed with Novocyte ACEA flow cytometer using FITC (530/30 BP) and mCherry (615/20 BP) filter cubes.

## RNA extraction and quantification

Total RNA was isolated from cells, dissociated and sorted tumours or frozen tissue samples using Rneasy mini kit (Qiagen). cDNA was prepared from 500 ng of total RNA using QuantiTect Reverse

**Table 2.** Flow cytometry antibodies.

| Antibodies Myeloid panel | Clone | Source | Fluorochrome |
|---|---|---|---|
| Anti-Ki67 | B56 | BD Biosciences | BV786 |
| Anti-MHCII | M5/114 | BD Biosciences | BV650 |
| Anti-Ly6C | HK1.4 | BioLegend | BV605 |
| Anti-CD11b | M1/70 | BD Biosciences | BV412 |
| Anti-F4/80 | BM8 | BioLegend | PE-Cy7 |
| Anti-CD11c | N418 | Invitrogen | PE-Cy5.5 |
| Anti-CD45 | 30-F11 | BioLegend | AF700 |
| Anti-Ly6G | 1A8 | BioLegend | APC |
| Anti-CD24 | M1/69 | eBioscience | PerCp-eFlour710 |
| Anti-CD115 | AFS98 | Biolegend | PE/Dazzle |
| Anti-Ceacam1 | CC1 | eBioscience | PE-Cy7 |

Transcription Kit (Qiagen). SYBR green (see *Table 1*) or Taqman-based Quantitative PCR (hprt: Roche universal probe library #95 AGTCCCAG); Ras: Roche universal probe library #88 (CATCCTCC); Ceacam1: (TCTCACAGAGCACAAACCCTCAGC) were performed on Roche LightCycler480.

## RNAseq analysis

RNAseq bank were prepared from, respectively, three and four independent biological samples for the cell lines and Venus-positive cells sorted from tumours. We used *Universal Plus mRNAseq* kit from NuGEN and sequenced on Illumina HiSeq 2500. After quality control the sequences were aligned to Mm9 mouse genome using TopHat2. Normalization and differential gene expression between pairs was performed with Deseq. Gene set enrichment analysis was done using pre-ranked method based on fold changed values (GSEAv.6.0) and analysed with C5 ontology gene sets (*Subramanian et al., 2005*). Immune cell type deconvolution was done with CIBERSORTx (https://cibersortx.stanford.edu/) based on RPKM counts that were compared with murine immune signatures from 10 cell types produced by *Chen et al., 2017*.

## Flow cytometry analyses

Mouse tumours were dissociated into single-cell suspension using the Mouse tumour dissociation kit (Miltenyi Biotec) and the gentleMACS Octo Dissociator following the manufacturer's instructions and as described in *Taranto et al., 2021*. The cell suspension was passed through a 70 μm cell strainer (Miltenyi), centrifuged at 300× *g* for 10 min at 4°C and washed three times in FACS buffer. GFP-negative stromal cells were isolated from cell suspensions with a FACS sorter ARIAIlu (Becton Dickinson) and frozen in foetal calf serum with 10% DMSO. Frozen samples were thawed under sterile conditions and single-cell preparations were incubated with anti-CD16/CD32 antibody (BD Biosciences) for 15 min and stained with the indicated antibodies following standard procedures. Samples were fixed with eBioscience fixation/permeabilization kit (Invitrogen) and Ki67 antibody was used for intracellular staining (see *Table 2*), and analysed as previously described (*Wang et al., 2019*). The

**Table 3.** IHC antibodies.

| Name | Clone | Source | Species | Cat # | Dilution |
|---|---|---|---|---|---|
| Ki67 | SP6 | Spring Bioscience | Rabbit | M3064 | 1:250 |
| Caspase 3 | ASP175 | Cell signalling | Rabbit | 9661S | 1:4000 |
| CD31 | Polyc | Abcam | Rabbit | Ab28364 | 1:75 |
| MFG-E8 | 18A2-G10 | MBL | Hamster | D199-3 | 1:1000 |

signal was detected by a 4-laser Fortessa flow cytometer (Becton Dickinson). Analyses were carried out using FlowJo software.

## Immunohistochemical analysis

Three μm sections of paraffin-embedded samples were analysed using a Ventana Discovery Ultra automated staining instrument (Ventana Medical Systems), according to the manufacturer's instructions. Briefly, slides were de-paraffinized, the epitope retrieval was performed with the reagents provided by the manufacturer, the endogenous peroxidase was blocked, and the samples were incubated with the appropriate antibodies for 60 min at 37°C (*Table 3*). Signal enhancement for caspase 3 and Ki67 antibodies was performed either with the OmniMap anti-rabbit detection kit or by the Discovery HQ-conjugated anti-rabbit IgG followed by Discovery amplification anti-HQ HRP Multimer, according to the manufacturer's instructions.

Slides were incubated with DAB chromogen and counterstained with hematoxylin II for 8 min, followed by Bluing reagent for 4 min. Brightfield stained slides were digitalized with a Hamamatsu NanoZoomer 2.0-HT scanner and images were visualized with the NDP view 1.2.47 software except for the MFG-E8 staining, for which slides were digitally processed using the Aperio ScanScope (Aperio). Nodule size was drawn by hand in HALO image analysis software (Indica Labs) and an algorithm was designed with the Multiplex IHC v1.2 module to quantify the percentage of positive cells per mouse, as indicated in figure legends.

## Immunofluorescence

Frozen sections were equilibrated at room temperature for 10 min and rehydrated in PBS. Sections were incubated overnight with anti-mouse NKp46/NRC1 antibody (1:500, AF2225 R&D Systems) and for 1 hr with rabbit anti-goat Alexa647-conjugated secondary antibody (Invitrogen A21446). Images were acquired using Zeiss axioimager Z1 microscope with DAPI, GFP, and Cy5 filter cubes.

## NK cells cytotoxicity assays

NK cells enrichment: for each independent experiment, spleen was harvested from an athymic nude mouse, mashed on a 70 μm cell strainer and NK cells were enriched using the MACS NK cell negative selection kit (#130-115-818, Miltenyi Biotec). The enriched NK cells populations were resuspended in RPMI 1640 medium supplemented with 10% FCS, 1% penicillin/streptomycin, sodium pyruvate (1 mM), HEPES (10 mM), and β-mercaptoethanol (50 μM), counted and used to set up cytotoxic assays. Efficiency of the enrichment was evaluated by flow cytometry using antibodies directed against CD45 (clone 30-F11, BD Biosciences), enrichment 90–95%; and Nkp46 (clone 29A1.4, Thermo Fisher), enrichment: 80–90%.

BMEL target cells (clones Ras$^{LOW-mCherry}$ and Ras$^{HIGH-Venus}$) were seeded at a 1:1 ratio in a 96-well plate (flat-bottom) pre-coated with collagen type I (BD Biosciences 354236) (in total, 20,000 target cells/well). Cells were incubated 2 hr at 37°C, then enriched NK cell suspensions were added at different ratios to the co-culture of target cells, as indicated (20,000–100,000 cells). The cytotoxicity test was performed in complete RPMI media in the presence of mIL15 (100 ng/ml, R&D Systems). Duplicates were generated for each condition tested. After 16 hr of incubation at 37°C, supernatant and cells were quickly collected by trypsinization and kept on ice. A viability dye was added extemporaneously (DAPI, 400–600 ng/ml final) and the percentage of DAPI+ cells in Ras$^{LOW-mCherry}$ and Ras$^{HIGH-Venus}$ cells was evaluated by flow cytometry. As a control (baseline level), the percentage of DAPI+ cells was measured in co-cultures of target cells maintained under the same media conditions but not exposed to NK cells.

## Softwares

The following open-source code and software were used in this study: R (v.3.5.1), Default settings were used for all GSEA. R package DESeq2 (v.1.22.1), R package heatmap (v.1.0.10), and R package VennDiagram (v.1.6.20) were also used. GraphPad Prism v8 was used to make graphs and for statistical analyses.

## Experimental study design and statistics

The investigators were blinded every time it was possible. Blinding was not possible when the investigators needed to have the sample identification in order to perform downstream analyses. Data

sets were tested with two-tailed unpaired or paired Student's t tests, one-sample t-tests, or two-way ANOVA using Prism Software version 8 (GraphPad). Significant p-values under 0.05 were considered as significant.

## Acknowledgements

This work was supported by the grant HTE from ITMO Cancer (to UH), from the Association Française pour l'Etude du Foie (AFEF, to DG) and from SIRIC Montpellier Cancer (to UH). AL received support from the University of Montpellier and from the Fondation ARC and SIRIC Montpellier Cancer. LA, SV, and CR are supported by the Dutch Cancer Society (KWF 12049/2018-2), Oncode Institute and the Center for Cancer Genomics (CGC.nl). We acknowledge the 'Réseau d'Histologie Expérimentale de Montpellier' – RHEM facility, supported by SIRIC Montpellier Cancer (Grant INCa_Inserm_DGOS_12553), the European regional development foundation and the Occitanie region (FEDER-FSE 2014–2020 Languedoc Roussillon), for the histological analyses. We acknowledge the MGX facility for the RNAseq analysis, and the MRI imaging platform. We thank the RAM animal facility and IGMM zootechnicians for care of animals. We are especially grateful to Myriam Boyer and Stéphanie Viala for help with cell sorting. We thank José Ursic-Bedoya for the cell line derived from the hydrodynamic gene transfer experiments and Benjamin Rivière for his expert advice on anatomopathology. We gratefully acknowledge helpful discussions with Eric Assenat and all other members of the laboratory. We are especially grateful to the direction of the IGMM institute for ensuring the conditions that allowed us to continue our work during the COVID-19 pandemic.

## Additional information

### Funding

| Funder | Grant reference number | Author |
|---|---|---|
| SIRIC Montpellier Cancer | Grant INCa_Inserm_ DGOS_12553 | Urszula Hibner |
| Grant HTE-ITMO Cancer | HTE201610 | Urszula Hibner |
| Association Francaise pour l'Etude du Foie | | Damien Grégoire |
| KWF Kankerbestrijding | KWF 12049/2018-2 | Leila Akkari |

The funders had no role in study design, data collection and interpretation, or the decision to submit the work for publication.

### Author contributions

Anthony Lozano, Conceptualization, Investigation, Visualization, Writing – review and editing; Francois-Régis Souche, Carine Chavey, Valérie Dardalhon, Christel Ramirez, Serena Vegna, Anaïs Riviere, Investigation; Guillaume Desandre, Formal analysis, Investigation, Visualization; Amal Zine El Aabidine, Formal analysis; Philippe Fort, Formal analysis, Writing – review and editing; Leila Akkari, Supervision, Funding acquisition, Writing – review and editing; Urszula Hibner, Conceptualization, Supervision, Funding acquisition, Writing – original draft, Writing – review and editing; Damien Grégoire, Conceptualization, Supervision, Funding acquisition, Investigation, Visualization, Writing – original draft, Writing – review and editing

### Author ORCIDs

Philippe Fort ⓘ http://orcid.org/0000-0001-5997-8722
Damien Grégoire ⓘ http://orcid.org/0000-0002-1105-8115

### Ethics

All reported animal procedures were carried out in accordance with the rules of the French Institutional Animal Care and Use Committee and European Community Council (2010/63/EU). Animal studies were approved by institutional ethical committee (Comite d'ethique en experimentation animale

Languedoc-Roussillon (#36)) and by the Ministere de l'Enseignement Superieur, de la Recherche et de l'Innovation (APAFIS#11196-2018090515538313v2).

### Decision letter and Author response

Decision letter https://doi.org/10.7554/eLife.76294.sa1
Author response https://doi.org/10.7554/eLife.76294.sa2

## Additional files

### Supplementary files

• Supplementary file 1. RNAseq and gene expression data. (a) Gene signature for the low threshold Ras/MAPK signalling (*Figure 1E* left panel). (b) Gene expression modified only in RasHIGH cells (*Figure 1E* middle panel). (c) Genes for which expression correlated with intensity of Ras/MAPK signalling (*Figure 1E* right panel). (d) Genes differentialy expressed in peritoneal tumour cells versus liver tumour cells. (e) GSEA identified enriched gene sets liver versus peritoneum isolated tumour cells.

• MDAR checklist

### Data availability

The RNA-sequencing data have been deposited in the Gene Expression Omnibus (GEO, NCBI) repository, and are accessible through GEO Series accession number GSE180580. Raw data from figures 1 to 5 were deposited on Mendeley data at doi: https://doi.org/10.17632/73nbvs8925.1.

The following previously published datasets were used:

| Author(s) | Year | Dataset title | Dataset URL | Database and Identifier |
|---|---|---|---|---|
| Lozano A, Hibner U, Gregoire D | 2021 | Transcriptomic analysis of mouse hepatic progenitors expressing different level of the HRasG12V oncogene (in vivo and ex vivo) | https://www.ncbi.nlm.nih.gov/geo/query/acc.cgi?acc=GSE180580 | NCBI Gene Expression Omnibus, GSE180580 |
| Grégoire D | 2021 | Ras/MAPK signalling intensity defines subclonal fitness in a mouse model of primary and metastatic hepatocellular carcinoma | https://doi.org/10.17632/73nbvs8925.1 | Mendeley Data, 10.17632/73nbvs8925.1 |

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
