## [Editor Report]

This is an important study that demonstrates a role for elevated Ras signaling in the growth of liver cancer in immunocompetent mice that is mediated, in part, by resistance to NK-mediated killing. The conclusions are, in large part, supported by solid data. Further studies are required to identify molecular mechanisms.

---

## [Decision Letter]

[Editors' note: this paper was reviewed by Review Commons.]

Thank you for submitting your article "Ras/MAPK signalling intensity defines subclonal fitness in a mouse model of primary and metastatic hepatocellular carcinoma" for consideration by eLife. Your article has been reviewed by 3 peer reviewers at Review Commons, and the evaluation at eLife has been overseen by a Reviewing Editor and Päivi Ojala as the Senior Editor.

Based on your manuscript and the reviews, we invite you to submit a revised version incorporating the revisions as outlined below:

This study was designed to investigate Ras-stimulated tumor formation by bipotential mouse embryonic liver cells. The authors examine the effect of Ras dose on tumor formation in the liver and peritoneum. The authors also compare tumor formation in immune-competent and immune-incompetent mouse models, but do not extend the analysis to an examination of Ras dose. The authors conclude that: (1) pre-existing tumor initiating cells with a specified Ras dosage accounts for tumor formation and that the Ras dosage requirement is different for the liver and peritoneum, (2) that the immune microenvironment differs between the liver and peritoneum due to the efficiency of antigen processing by dendritic cells and the presence of NK cells, and (3) that strong MAPK signaling is required primary liver tumors to resist elimination by NK cells.

Data to support these conclusions is missing from the study presented and further experimentation is required to address these issues. The following specific revisions are requested.

1) The authors frequently refer to metastasis in the text of the manuscript, including in the abstract. However, the transplantation model is not a metastatic model of cancer. Indeed, the authors state that the location of peritoneal cells occurs at the time of injection. The text needs to be corrected throughout.

2) The authors conclude that there is selection for pre-existing clones with a specific level of Ras expression for the growth of liver and peritoneal tumors. This seems likely, but not directly demonstrated. If Ras expression is variable (epigenetics etc), is it possible that selection occurs at the level of Ras expression? Studies with single clones would be helpful.

3) Studies using immunocompetent mice transplanted with liver tumor cells are presented (Fig. 4G). The earlier nude mouse studies were performed using Ras-transformed BMEL cells. Is the absence of peritoneal tumors in the immunocompetent mice because liver tumor cells were employed rather than Ras-transformed BMEL? A different experimental design would be to use syngeneic BMEL cells for all of these studies. How do the authors justify the presented experimental design that prevents comparison of the immunocompromised and immunocompetent mice.

4) The studies of immunocompetent mice (Fig. 4G) does not address Ras dosage. Does the level of Ras expression affect tumor formation (like nude mice)? Is there selection for the level of Ras expression (like nude mice)? These questions are not addressed.

5) The abstract refers to an "evolutionary trade-off of increased apoptotic death in the liver", but no data on increased apoptotic death of RasHigh cells in the liver is presented.

6) A role of NK cells is inferred from the comparison of nude mice with high levels of NK cells and NSG mice with few NK cells. Differences in Casp3+ cells are described, but the tumor volumes were comparable. The role of NK cells was not directly tested. The authors' conclusions are therefore unclear.

7) A major conclusion drawn from this study is that strong MAPK activation is required for primary liver tumor resistance to NK cell-mediated elimination (stated in abstract). The only experiment presented that proposes to address a role for NKT cells is the NSG mouse transplantation study, but this study showed comparable liver tumor volumes with nude mice. How is this consistent with the stated conclusion?

8) The abstract states that MAPK Erk signaling intensity contributes to potential vulnerability in HCC. This should be re-stated as Ras rather than Erk because Erk represents only one the Ras effectors and the relative roles of different Ras effectors was not tested.

---

## [Author Response]

Reviewer 1:While studies are very elegant and results convincing, it is unclear how they might be deployed to therapeutic ends.Reviewer 3:The authors should explain why this is an interesting finding. They mention in the abstract that this heterogeneity highlights potential vulnerabilities that could be therapeutically exploited. How do they envision this? Why is this not a trivial result and in what way can this observation help design new therapies?

We believe that our results suggest exciting opportunities in the search for novel therapeutic options and we agree further discussion on this important issue should be included in the revised manuscript. We have now expanded the discussion on these points and commented on the clinical relevance of our findings to answer the reviewers' concern (Discussion section page 15 lines 7-15).

Our data are consistent with the widely acknowledged role of NK cells in anti-tumour immunity (e.g. Pende et al., Frontiers Immunol. 2019, 10:1179, for review). NK activity is governed by engagement of a repertoire of activating and inhibitory receptors expressed on their surface (Shimasaki et al., Nature Rev Drug Discovery, 2020, 19, 200-218). Among the latter, homophilic interactions of CEACAM1 (CD66A) expressed on melanoma cells have been shown to protect the tumor from NK-mediated toxicity (Markel et al., J Immunol 2002; 168:2803-2810;), in a strict parallel to our interpretation.

NK cells are relatively sparse in the peripheral blood and abundant in a healthy liver. In patients with HCC, the numbers of peripheral, liver resident and tumor-infiltrating NK all drop significantly, mainly due to the disappearance of CD56^dim^CD16^pos^ cell subset, corresponding to the cytotoxic NK population. Moreover, despite the continuous expression of activating receptors, the functionality of both the cytotoxic (and the cytokine producing (CD56^bright^CD16^neg^)) remaining NKs is severely impaired (Cai et al. Clinical Immunology (2008) 129, 428–437). The molecular mechanisms underlying NK anergy in the context of HCC have yet to be fully elucidated. However, CEACAM1 expression has been shown to suppress NK function in hepatitis C patients (Suda et al. Hepatology Communications 2018;2:1247-1258) and there is ample evidence of CEACAM1 playing a major role in hepatic disease and in particular in protection against inflammation and immune-induced hepatitis (reviewed in Horst et al. Int. J. Mol. Sci. 2018, 19, 3110). Thus, CEACAM1 is a bona fide regulator of NK function that is relevant in cancer and in non-cancerous liver pathology.

In this context, our data introduce an additional notion, namely the tumor-promoting effect of a strong ERK activation in HCC that leads to CEACAM1-mediated anergy of NK.

How might these findings be translated into future therapeutic options for HCC? Several scenarios can be envisaged, a very attractive being a cell-mediated immunotherapy, notably either autologous or allogeneic NK transfer. These therapies, which were initially developed for hematopoietic malignancies, are currently gaining momentum for solid tumors. Infusion of modified NK cells, including CAR-NK, presents major advantages over T-cell based therapies, mainly due to a very much diminished risk of GVDH in allogenic setting and of cytokine release syndrome and neurotoxicity for autologous transfer. Moreover, because NK-mediated cytotoxicity is HLA-independent, it does not require careful haplotype matching, thus greatly increasing the speed and availability of cellular preparations (recently reviewed in Xie et al. EBioMedicine 59 (2020) 102975).

Currently, there are 219 registered clinical trials, including 31 on HCC, for NK-mediated anti- solid tumor responses (clinicaltrials.gov). Although most of these are only in phase I or phase II, they bear a great promise for the future. Our data strongly suggest that a new combination therapy might have an improved efficiency in a subset of HCC characterized by a strong ERK activation. This would involve either activated NK or CAR-NK in combination with a FDA-approved inhibitor of the MAPK ERK, such as trametinib. Our data lead us to predict that even a partial decrease in the intensity of ERK signaling would be likely to significantly increase the efficacy of NK-mediated anti-tumor activity, at least in a subset of HCC. While we appreciate that this suggestion remains speculative at this point in time, we believe the strength and novelty of our data warrants an exploration of such novel therapeutic opportunity for this tumor type that dramatically lacks reliable treatment options.

Specific pointsReviewer 1:Minor edits: Editorial review for minor, infrequent word usage edits

We apologize for any English language mistakes in the manuscript. While the formulation of the remark makes us believe that our word usage does not impair the understanding of the text, we shall of course be willing to correct it.

Figure 1E: Not possible to read genes in left heatmap, middle heatmap very small. Figure 3D: Units at x and y axes not legible/small. 3E: scales not legible. Figure 4: typo in legend H-> G.

We apologize for not being more careful in preparing these figures, this has now been corrected. We realize that due to the high number or genes, their names are still in a very small print in the left panel of Figure 1E, however, the complete list the genes is given in the supplementary table 1, and we added larger image of the heat map with the table.

Reviewer 2:1. Is there any significant change of EMT like status in BMEL cells having H-RAS (high) vs H-RAS (low)?

Several EMT markers (e.g. vimentin or loss of E-cadherin) are induced by H-Ras in BMEL cells, as we have previously reported (Akkari et al. J Hepatol, 2012). Moderate levels of Ras expression appear to be sufficient for this phenotype, since we did not detect significant differences in their expression profiles either between RAS^HIGH^ and RAS^LOW^ populations or between cells isolated from the hepatic versus the peritoneal tumors. We conclude that the phenotype of a selective advantage afforded by a high RAS expression level is not due to the EMT.

2. There is no significant fold difference (MFI number) to put the sorting gates to enrich H-RAS high vs H-RAS low cells. The mRNA expression level was almost 3 fold difference. Is it correlated with protein expression level?

This is a very valid point. We agree that the MFI difference is not strong, although it is in fact statistically significant in three independent cell sorting experiments. We were confident that the differences in the H-Ras mRNA level were reflected in the level of protein expression, since we have observed distinct transcriptional signatures as well as significant phenotypic differences in Ras^HIGH^ and Ras^LOW^ cells (Figure 1 B and D). Nevertheless, we quite agree that the difference in protein expression level needed to be confirmed. This has now been done by immunoblot analysis of protein extracts with an antibody specific to Ras^G12V^ (Cell signaling #14412). These data have now been included in Figure 1A, and the text modified accordingly (page 4 line 9-19).

3. Is there any translational relevance of these genes Al467606, Aim2, Dynap, Htra3, Itgb7, Tspan13 in HCC patients with poor survivability?

The expression of these genes positively correlated with the level of the Ras oncogene in the ex vivo cell culture model, thus providing a nice demonstration that variation in HRAS oncogenic dosage translates into differential transcriptomic outputs. The analysis of publicly available data from the cancer genome atlas (TCGA) also showed their expression in the HCC cohort (372 patients samples).

**Author response image 1. sa2fig1:** 

The clinical outcome of the level of their expression (see Author response image 1) is somewhat ambiguous: strong expression of ITGB7 and C16ORF54 (Human ortholog of Al467606) correlated with a better prognosis, while expression of AIM2 and DYNAP had no impact on patient overall survival. Finally, HTRA3 and TSPAN13 were associated with worse outcomes and thus constitute particularly interesting candidates for future investigations. These somewhat unexpected divergent correlations likely reflect the fact that RAS/MAPK signalling is unlikely to be the sole regulator of their expression.

4. Is there any difference between survival curve upon grafting of H-RAS (high) vs H-RAS (low) cells in Figure 2A?

This experiment has not been performed for ethical reasons. Indeed, the difference in tumor growth upon injection of Ras^HIGH^ vs Ras^LOW^ is statistically significant 21 days after injection (Figure 2A, p-value = 0.008). The size of the Ras^HIGH^ tumours is rather large and we chose to sacrifice the animals before they developed any signs of suffering.

5. Is there any difference of H-RAS expression between liver tumor and peritoneal tumors?

We have quantified H-Ras expression levels by RTqPCR in the flow cytometry sorted tumoral cells derived from the liver and peritoneal tumours (Figure 3C). In the revised version of the manuscript we provide evidence that the mRNA expression levels of the oncogene correlate with the protein expression. Therefore, while the measurement of H-Ras protein has not been performed on the tumours, we would argue that it will indeed be different in the two tumour locations.

6. Please provide the data for pro-inflammatory cytokines in TME.

These data have been shown in the Suppl. Figure 4C.

7. Please provide an explanation of the DC activation with antigen presentation though the tumor is non-necrotic or apoptotic.

While it is true that peritoneal tumors are less necrotic and have a lower apoptotic index than the matched hepatic primary ones (Figure 3E), significant cell death can be detected at both locations. We assume that the released antigens are sufficient for presentation by the DC, as supported by the data in Figure 4C, D and G.

8. Is the TAM showing M2 phenotypes at peritoneal tumors?

The reviewer correctly points out that the distinction between liver and peritoneal TAM polarization is not perfectly clear-cut, since some immunosuppressive but also some inflammatory markers are present at both tumor locations (Figure 4B and Suppl Figure 4). This is not unexpected, as the spectrum of activation macrophages can undertake in vivo is neither static nor fully faithful to the M1/M2 polarization extremes inducible in vitro (see e.g. Ringelhan et al., Nat Immunol. 2018;19(3):222-232 ; Ruffell et al., Trends Immunol. 2012 33(3):119-26). We thus integrate these results with our observations of other modulated immune cell phenotypes in these tumors. Indeed, in addition to the macrophage polarization markers, we noted a more mature, activated phenotype in the peritoneal TAMs. Together with the cytokine expression profile in the two tumor locations (which is included as a supplementary table in the revised version of the manuscript) our data argue for a less inflammatory environment in the peritoneal tumors.

SignificanceThe data showed pretty promising and has a seminal impact on H-RAS high expressing HCC patients. TAM and DC showed some important immune regulation to promote HCC.

We thank the reviewer for his appreciation of the significance of our study.

Reviewer 3:It is possible that RAS levels may not stay constant but dynamically go up and down. While this is a possibility that would complicate interpretations of the results, I am ok with the conclusions in the manuscript as it is, since there seems to be a significant difference between the different populations assayed.

This is a valid point that we have addressed by comparing the H-RAS expression level in the parental BMEL population (labelled “cells before injection” in Figure 2D) to those either freshly isolated from the tumors after a rapid cell-sorting by flow cytometry (“tumors” in Figure 2D) and then to those isolated from tumors and kept in culture for 14 days (“tumoral cell lines” in Figure 2D). Our conclusion was that the level of RAS expression was stable upon ex vivo culture. This result does not exclude a possibility of epigenetic regulation that operated in vivo and was maintained in the subsequent cell culture. However, even if this was the case, it would not alter the conclusion of distinct selective advantage of the HRAS expression levels in the two tumoral locations.

SignificanceThe authors should explain why this is an interesting finding. They mention in the abstract that this heterogeneity highlights potential vulnerabilities that could be therapeutically exploited. How do they envision this? Why is this not a trivial result and in what way can this observation help design new therapies?

This important point is very similar to the concern raised by the reviewer 1 and we have answered them together at the beginning of the rebuttal.

We would like to thank again the reviewers for raising this issue, which prompted us to include the considerations of potential usefulness of our findings in the revised discussion.

[Editors' note: further revisions were suggested prior to acceptance, as described below.]

1) The authors frequently refer to metastasis in the text of the manuscript, including in the abstract. However, the transplantation model is not a metastatic model of cancer. Indeed, the authors state that the location of peritoneal cells occurs at the time of injection. The text needs to be corrected throughout.

We agree that the peritoneal tumours in our experimental model are probably not *bona fide* metastases, since they are likely to be seeded during the orthotopic injection. Nevertheless, our experimental model allows the examination of hepatic tumour growth at the primary (liver) site and a location known to support HCC metastasis in human patients (the peritoneum). We have therefore corrected the text, keeping the terms "metastatic site" and "metastatic-like tumours", but removing "metastasis" and "metastatic tumours" as such. We have also removed the term of “metastatic” from the title of the article.

2) The authors conclude that there is selection for pre-existing clones with a specific level of Ras expression for the growth of liver and peritoneal tumors. This seems likely, but not directly demonstrated. If Ras expression is variable (epigenetics etc), is it possible that selection occurs at the level of Ras expression? Studies with single clones would be helpful.

This is a very valid point. Indeed, our data do not exclude the possibility of variable Ras expression that could arise via epigenetic regulation under different growth conditions. Data presented in Fig 2D argue against this interpretation. Moreover, we are working with populations of cells harbouring different copy numbers of the oncogene inserted into different genomic loci. Therefore, in a scenario of epigenetically-driven regulation of oncogene expression, one would expect some, but not all of the subclones to be susceptible to upregulate Ras expression in response to the in vivo microenvironment. This would give rise to a population of cells with a range of Ras expression levels that would be different from the original population that was allografted, but would remain subject to selection operating in vivo. We therefore believe that epigenetic regulation of Ras expression might indeed exist, but is unlikely to account for the observed phenotype.

Regarding the use of single clones: we chose to use cell populations since they are best adapted to study selection processes operating in vivo. We have nevertheless set out to isolate clonal cell lines from the BMEL-Ras populations. The majority of the single clones analysed (n=26) expressed low to moderate Ras levels. We have verified the stability of Ras expression levels in culture and used Ras^HIGH^ and Ras^LOW^ clones in the NK cytotoxicity assays described below.

3) Studies using immunocompetent mice transplanted with liver tumor cells are presented (Fig. 4G). The earlier nude mouse studies were performed using Ras-transformed BMEL cells. Is the absence of peritoneal tumors in the immunocompetent mice because liver tumor cells were employed rather than Ras-transformed BMEL? A different experimental design would be to use syngeneic BMEL cells for all of these studies. How do the authors justify the presented experimental design that prevents comparison of the immunocompromised and immunocompetent mice.

The BMEL clone we have used in this work was chosen because of its very high oncogenic potential: unpublished data from our lab indicates that <500 cells are sufficient for orthotopic tumour formation. However, the cells are of a mixed genetic background, making them unsuitable for syngeneic transfer. Therefore, for testing the behaviour of Ras-transformed cells in immunocompetent animals we have turned to an alternative model of a cell line derived from mouse tumours initiated by hydrodynamic gene transfer. We quite agree that the results are not directly transposable to the model of *ex-vivo* transformed BMEL cells, but we disagree with the statement that the experimental design prevents comparison of the immunocompromised and immunocompetent mice. Indeed, the same tumour-derived cells were injected into wt B6 (syngeneic) animals and into nude mice (Fig. 4G). As expected, peritoneal tumours were observed in immunodeficient, but not immunocompetent mice, arguing for an efficient immune control in the peritoneum, consistent with our data suggesting differences in antigen presentation capacity at the two tumoral sites.

4) The studies of immunocompetent mice (Fig. 4G) does not address Ras dosage. Does the level of Ras expression affect tumor formation (like nude mice)? Is there selection for the level of Ras expression (like nude mice)? These questions are not addressed.

The experiments presented in Fig. 4G were not designed to investigate the Ras dosage effect in immunocompetent host, but to test the prediction that efficient antigen presentation in the peritoneum would prevent tumour establishment at this location in the mice capable of mounting an immune response.

While we did measure Ras dosage in the B6 liver tumours, which was not significantly different from the ex vivo grown cells, we did not further investigate this point. This is because for this experiment we used cells that are invalidated for the expression of the TP53 tumour suppressor, a condition necessary to obtain Ras-driven tumours by hydrodynamic gene transfer. The same cells were used for assays in C57Bl/6 and nude mice (Fig. 4G), allowing a direct comparison of tumours arising in immunocompetent and immunodeficient animals. We have however not investigated further the Ras dosage dependence in the context of inactivated p53.

5) The abstract refers to an "evolutionary trade-off of increased apoptotic death in the liver", but no data on increased apoptotic death of Ras^High^ cells in the liver is presented.

Our data show that apoptosis, revealed by active caspase 3 and by the analysis of the RNAseq data, is correlated with high-intensity Ras signalling (Fig. 3D and E). Moreover, our unpublished data indicates that Ras^High^ cells have low clonogenicity. The simplest interpretation of these findings is that strong Ras signalling is deleterious to the transformed cells, in accordance with abundant data in the literature. However, we agree that the concept of the evolutionary trade-off remains an interpretation and not an experimental result. In consequence, we propose to add the word "apparent" in the summary ("significantly higher Ras expression was observed in primary as compared to secondary, metastatic sites, despite the apparent evolutionary trade-off of increased apoptotic death in the liver that correlated with high Ras dosage"). In the discussion, we say that our data "suggests an evolutionary trade-off for the high oncogene dosage in the liver tumours", clearly indicating that we use this concept to interpret our results.

6) A role of NK cells is inferred from the comparison of nude mice with high levels of NK cells and NSG mice with few NK cells. Differences in Casp3+ cells are described, but the tumor volumes were comparable. The role of NK cells was not directly tested. The authors' conclusions are therefore unclear.

We agree that a direct test of NK toxicity was missing from the previous version of the manuscript. We have now performed these experiments and the revised version of the manuscript includes data showing that Ras^HIGH^ cells are more resistant to NK-mediated toxicity than their Ras^LOW^ counterparts. These new results, presented in the revised Fig. 5C, further support our contention of the role of NK cells in progressive elimination of the Ras^LOW^ clones in liver tumours.

7) A major conclusion drawn from this study is that strong MAPK activation is required for primary liver tumor resistance to NK cell-mediated elimination (stated in abstract). The only experiment presented that proposes to address a role for NKT cells is the NSG mouse transplantation study, but this study showed comparable liver tumor volumes with nude mice. How is this consistent with the stated conclusion?

We acknowledge that the scarcity of experiments directly addressing the role of NK cells was a weakness of the previous version of the manuscript. The revised version now contains data from NK cytotoxicity assays showing increased resistance of Ras^HIGH^ compared to Ras^LOW^ clones. These data support our conclusion of the role of NK cells shaping the dynamics of tumour growth. Of note, we do not claim that the NKs are the sole factor responsible for differences in the liver and peritoneal tumours, in fact we do not believe this is likely to be the case. Indeed, while our data indicate NK participation in creating distinct selective pressures in the two tumour locations, molecular differences between the liver and peritoneal tumours are not totally abrogated in the NSG animals (see e.g. Fig. 5E).

Regarding the tumour volumes in the nude and NSG mice: the data in Fig. 5C show the relative ratio of peritoneal to liver tumours for each individual mouse rather than absolute tumour volumes. The reason for opting for such presentation is that tumour volumes vary considerably in different experiments, and also between animals in the same experiment (illustrated in Fig. 3B). In order to obtain statistically significant results, a large number of animals must be used and the price of the NSG mice was prohibitive for such a study. Calculating the ratio of tumour volumes at the two locations in the same animal was more informative than representation of absolute tumour sizes. Admittedly, it also suffers from the shortcoming of a low number of animals, which precludes reaching statistical significance of the data.

8) The abstract states that MAPK Erk signaling intensity contributes to potential vulnerability in HCC. This should be re-stated as Ras rather than Erk because Erk represents only one the Ras effectors and the relative roles of different Ras effectors was not tested.

It is of course true that MAPK Erk is not a sole effector of Ras and indeed we have not studied the role of alternative effectors in the tumour phenotype we describe. Interestingly, short-circuiting RTK engagement by direct manipulation of Ras activity has been reported to specifically trigger of the MAPK and not the PI(3)K/mTOR pathway (Toettcher et al. 2013, Cell, 155, 1422–1434. doi:10.1016/j.cell.2013.11.004). In our model neither the phosphoproteomic array (Fig. 1C) nor the transcriptomic analysis gave any indication of differential activation of the PI(3)K/Akt /mTOR pathway. This is in contrast to significant differences in Erk activity between Ras^HIGH^ and Ras^LOW^ BMEL cells, as well as cells isolated from hepatic versus peritoneal tumours, revealed both by transcriptomic and phosphoproteomic data (Figs 1C, 1D and 3D).

We acknowledge, however, that we have not formally proven that Erk signalling intensity is the Achilles' heel of a subtype of HCC. This is why we mention only a "potential" vulnerability. To further downplay our assertion, we have now changed the text from "highlights a potential vulnerability of a subtype of hepatocellular carcinoma as a function of MAPK Erk signalling intensity" to "points to a potential vulnerability of a subtype of hepatocellular carcinoma as a function of MAPK Erk signalling intensity".